# Wattpad as a resource for literary studies. Quantitative and qualitative examples of the importance of digital social reading and readers' comments in the margins

**Federico Pianzola**[1,2]☯*, **Simone Rebora**[3,4]☯, **Gerhard Lauer**[4]

**1** Department of Human Sciences for Education "R. Massa", University of Milano-Bicocca, Milan, Italy,
**2** School of Media, Arts and Science, Sogang University, Seoul, South Korea, **3** Department of Foreign Languages and Literatures, University of Verona, Verona, Italy, **4** Digital Humanities Lab, University of Basel, Basel, Switzerland

☯ These authors contributed equally to this work.
* federico.pianzola@unimib.it

**Data Availability Statement:** All legally sharable data files are available from the OSF database

## Abstract

The end of deep reading is a commonplace in public debates, whenever societies talk about youth, books, and the digital age. In contrast to this, we show for the first time and in detail, how intensively young readers write and comment literary texts at an unprecedented scale. We present several analyses of how fiction is transmitted through the social reading platform Wattpad, one of the largest platforms for user-generated stories, including novels, fanfiction, humour, classics, and poetry. By mixed quantitative and qualitative methods and scalable reading we scrutinise texts and comments on Wattpad, what themes are preferred in 13 languages, what role does genre play for readers behaviour, and what kind of emotional engagement is prevalent when young readers share stories. Our results point out the rise of a global reading culture in youth reading besides national preferences for certain topics and genres, patterns of reading engagement, aesthetic values and social interaction. When reading Teen Fiction social-bonding (affective interaction) is prevalent, when reading Classics social-cognitive interaction (collective intelligence) is prevalent. An educational outcome suggests that readers who engage in Teen Fiction learn to read Classics and to judge books not only in direct emotional response to character's behaviour, but focusing more on contextualised interpretation of the text.

## 1. Introduction

There are at least 30 million books almost always ignored when talking about how and how much people read nowadays. This bibliographic treasure is a fast and steadily growing collection of short stories, novels, and poems published on the online platform Wattpad. As a comparison, the US Library of Congress—one of the biggest libraries in the world—has around 39 million catalogued books and "nonclassified prints", and more than 72 million manuscripts,

"Wattpad Titles Corpus" (https://doi.org/10.17605/OSF.IO/5GXMN). Data of the Teen Fiction corpus cannot be shared publicly because the copyright is held by the authors. In accordance to the European GDPR, data of the comments corpus cannot be shared publicly because this will expose underage users to possible identification. The code (R markdown file) needed to reproduce the results is available as Supporting Information (S1 File) and at the following link: https://doi.org/10.5281/zenodo.3375654 A guide to access the data in the same way the authors did is available at the following link: https://github.com/SimoneRebora/RSelenium_scraping_tutorial.

**Funding:** FP received funding from the European Union's Horizon 2020 research and innovation programme under grant agreement No 792849, "Reading Literature in a Digital Culture". GL received funding from the Swiss National Science Foundation. The funders had no role in study design, data collection and analysis, decision to publish, or preparation of the manuscript.

**Competing interests:** The authors have declared that no competing interests exist.

collected over 200 years of activity [1]. Wattpad was founded in 2007. In short, there will soon be more books in computers and digital shelves than in our material literary collections.

If literary studies want to understand the reading culture of the youngest generations in the 21st century, we need to consider the phenomenon of digital social reading [2,3]. This is because of two reasons: first, there is a huge volume of reading happening on digital social platforms like Wattpad on a worldwide scale; secondly, users' comments in the margins can be an extremely valuable resource for empirically studying readers' responses [4]. On Wattpad, readers share their thoughts and emotional reactions to specific paragraphs, not just to the whole book as in the case of online reviews. This means that researchers can access detailed data about readers' response: comments that readers write while they are reading, briefly (and spontaneously) interrupting their activity before continuing it. It is a kind of thinking-aloud or real-time data [5,6], which has never been available on a scale of millions of readers. They "represent the actual responses of actual readers" [7]. Notes written on e-readers like the Amazon Kindle have a similar function and are an example of how the social affordance of digital media strongly affect the reading experience: many people use Kindle notes to engage in conversations unrelated to the book in which they appear [4].

Research on reader response has started to show interest in online reading groups and book reviews to test theoretical hypotheses, complement stylistic analysis, and as a source for ethnographic inquiry [8–10]. A striking difference from first-generation reader response theory concerns the concept of reader: works of critics like Jauss, Iser, and Eco focused mainly on abstractions and non-testable constructions—like *Erwartungshorizont*—rather than on actual readers [11–13]. Empirically-driven reader response studies, on the other hand, bring back the actual reader at the centre of the analysis, by recognising the relevance of their responses for the study of literature [14]. With a phenomenon like Wattpad, we also have the opportunity to bring readers into computational literary studies and cultural analytics, which usually had to divert towards publishing history, when trying to reconstruct the reception of books [15]. Moreover, so far databases of reader responses did not offer a large amount of data, e.g. the UK Reading Experience Database (RED) hosts about 30,000 entries [16]: a precious trove for reading historians, but quite modest when compared to the millions of comments available on Wattpad.

An additional advantage of studying reader response on Wattpad is that the ecological validity of the experimental method is not affected by the researcher who interrupts the reading experience or manipulates the text [17]. It is a "naturalistic study of reading" [18]. In a historical perspective, glosses in ancient manuscripts and marginalia on printed books can offer a kind of insight similar to Wattpad's comments, but their amount, cultural diversity, and circulation is very limited [7,19] compared to the extent of the social interactions enabled by an online medium. With about 70 million readers, Wattpad publishes in more than 50 languages and nearly 300,000 writers from 35 countries take part every year in the largest writing competition [20].

Wattpad is an example of how the circulation of literature has changed due to digital technology, leading to a literary landscape in which the transmission of stories happens through complex sociotechnical digital networks [21–24]. Although it has been operating since more than a decade, so far there are only a few studies exploring the social dynamics of authors and readers on Wattpad [25–31], and even fewer are focusing on the texts and comments [32–34]. A similar phenomenon—in terms of popularity among readers and autonomy from the publishing industry—is that of fanfiction. Fanfiction has received greater attention but has been mostly studied with qualitative approaches [35–38], which did not aim at offering a large-scale description of its variety and circulation. Among the few quantitative investigations of fanfiction online platforms, interesting results have been achieved by Milli and Bamman [39],

showing how fan-fiction deprioritizes main protagonists in comparison to canonical texts, and by Yin et al. [40], who created and analysed metadata to identify the most popular fanfiction genres. However, these cultural domains are understudied by researchers in comparison to their pivotal role among reading practices in today's society.

Our aim is broader than what has been achieved by already existing research. With this paper we both contribute to research on the social aspect of reading and advocate for the role of distant reading in contemporary literary studies. Digital methods in literary studies have been deemed effective mostly for literary history and large-scale investigations [41–44], although there are also examples of computer-assisted close reading [45]. We present several analyses of how quantitative methods can be used to study reader response, namely using Wattpad to understand how fiction is read nowadays and to gain unprecedented insight into readers' interaction with texts and with other readers. Recently, Rowberry [46] has shown how even the most sophisticated and pervasive sociotechnical reading ecosystems like Amazon are currently very limited for the study of reading practices. However, our contribution presents a methodology that successfully uses big data to understand the new world of reading and writing literature.

We explored both the social and the textual dimensions for books written in various languages and spread worldwide. Our analyses are guided by three hypotheses: 1. analysing the themes of stories published on Wattpad we can highlight differences and similarities of reading interests across different languages, contributing to the study of world literature; 2. analysing users' behaviours with different genres we can show that readers' engagement changes according to the literary prestige of texts; 3. comparing the emotional valence of stories and comments we can provide empirical evidence to link textual features to readers' emotional response.

To explore the research aims, we used digital mixed methods and scalable reading [47–49], presenting analyses of different scope, and showing how quantitative analysis can be fruitfully used to identify parts of text for subsequent qualitative analysis, including close reading of excerpts from novels. Starting from describing the themes of the stories published on Wattpad (sections 2.2 and 3.2), we then move to mapping the geographical distribution of readers (sections 2.3 and 3.3). The next step is to show the difference between born-digital stories (originally published on Wattpad) and institutionalized literature (Classics), quantitatively describing readers' engagement with the two categories (sections 2.4 and 3.4). Then we show an original application of computational methods for text analysis to empirically study reader responses to literary texts, namely focusing on two novels. Guided by computational results, we qualitatively analysed the content of comments and the readers' interactions in relation to the text paragraphs that prompted them (sections 2.5 and 3.5).

## 2. Methods

To show how Wattpad can be used as a resource for understanding literary dynamics in the 21st century and for investigating reader response, we designed our research combining quantitative and qualitative analysis in a mixed methods approach suitable for computer-assisted literary studies [49]. One of the main characteristics of this approach is *scalable reading*, that is the combination of natural language processing and hermeneutics to "zoom in and out of things and discover that different properties of phenomena are revealed by looking at them from different distances" [47].

Perhaps *literary modelling* is a better term than *scalable reading*, since it does not suggest a practice limited to reading texts and stresses more the need to make the assumptions explicit about the nature of reading (literary) texts [50,51]. This includes a quantitative sociological

investigation of reading behaviours. However, recent advocates of literary modelling use the term mainly with respect to its application to literary history [43,44]. We agree that it is a good term that can help position distant reading in a constructive dialogue with close reading, but we also propose that this inclusive effort should be wider, aiming at taking into account other computational methods that work with literary texts but do not engage directly with literary history. Reader response studies can surely debate the historical dimension of reading practices, but they also synchronically investigate the emotional and cognitive aspects of reading [12,14,52].

Adopting this perspective, we tested hypotheses, explored the results generated from data analysis looking for large scale patterns, and selected case studies to close read, in order to understand how the identified patterns emerge from smaller scale phenomena. In other words, we started from the relationship between readers and text, and only then began to look for textual features. For an example of an approach starting from textual features to determine their "emotion potential", see Jacobs [53].

Our investigation can be divided into five groups of reflections. 1. The extent of the Wattpad phenomenon, i.e. how many stories are published, what are the different languages, and how many stories per language there are. 2. The most popular themes, including differences and similarities between stories written in different languages. 3. Stories' worldwide reception based on readers' comments. 4. Readers' engagement in terms of number of written comments and interaction with other readers. 5. The relations between text and reader's emotional response, confronting Teen Fiction and Classics.

In the following subsections we explain in detail the methods we applied and the procedures for the analyses.

## 2.1 Corpus building

Wattpad is mainly a community of readers and emerging authors who can read and publish texts via website or mobile app. It hosts a great variety of editorial forms and genres, but there is one crucial distinction that is of interest for literary scholars: most of the stories are born-digital, i.e. published for the first time on Wattpad and often written on smartphones [54]. However, there is also a small group of literary Classics, either imported from Project Gutenberg or published directly by Wattpad [55,56]. We did not perform a preliminary selection of the corpus. Instead, we collected data for all the stories indexed in Wattpad's sitemap. We then defined a smaller corpus, based on language popularity, for which to gather progressively higher density of data and analyse themes and reader response. Another reduction of scale brought us to define an even smaller corpus of Teen Fiction (TF) and Classics, categories that can be considered as representative of popular and prestigious literature respectively, according to common opinions in literary studies [44,57]. It is worth noting that the Classics category existed in January 2018 but it doesn't exist anymore in the website's frontend (November 2019), although all the books are still available on Wattpad and the category can be accessed directly typing the URL https://www.wattpad.com/stories/classics. However, the reliability of the current categorization is now compromised, displaying all sort of titles. Overall, we collected data about the number of reads, votes and comments for 1,000 titles in each category. Then we collected the number of comments to each paragraph for the 20 most read stories in each of the two categories, and the content of comments for 6 titles in each category.

To build the corpus of titles, we downloaded all links to Wattpad stories listed in the Wattpad sitemap (dated January 2018). The links are in a subsection of the sitemap, comprised between http://s3.amazonaws.com/wattpad/sitemaps/2018-01-23/sitemap1.xml and http://s3. amazonaws.com/wattpad/sitemaps/2018-01-23/sitemap144439.xml. Story titles were extracted

directly from the links, by decoding all special characters, substituting dashes with single spaces, and removing unique identifiers (e.g. "https://www.wattpad.com/story/21-about-my-character%27s" was transformed into "about my character's").

The dataset for TF and Classics stories has been expanded from a previous study [58]. It was divided into three sub-corpora: general metadata, fine-grained metadata, and full text/comments. The general metadata corpus hosts a total of 2,000 titles (1,000 for category), with information on the number of reads, votes, and the tags assigned by users. The fine-grained metadata corpus hosts a total of 300 titles (145 for the Classics, 155 for TF), selected from the general metadata corpus through an "importance" score that multiplies number of reads and of votes per title. This sub-corpus provides more detailed information on the number of reads, votes, and comments per chapter, together with the distribution of comments per paragraph. Finally, the full text/comments corpus hosts a total of 12 titles, selected among the most commented for each category (6 for Classics, 6 for TF). This corpus is structured as a table, with a unique identifier for commented book, chapter, and paragraph, the full text of comment and commented paragraph, and metadata on the comment (name of the commentator, date of publication, indication if the comment is a reply to a previous comment). All data was collected between January and July 2018, through a scraping algorithm written in R language and developed on the Remote WebDriver Selenium 2.0 (based on the Docker platform). For more information on the algorithm and on the corpus structure, see Rebora and Pianzola [34].

We informed Wattpad Corp. of our intention to scrape the website for research purposes and they did not object, pointing out that we needed to consider that users own the copyright of stories and comments. After consulting with the legal offices and the data protection boards of the institutions with which we are affiliated, we proceeded to collect the data implementing procedures to protect the users' identity. All usernames have been anonymised before analysis and the dataset of comments has not been shared publicly. Regarding TF stories' copyright, all procedures have been carried out through servers hosted in Germany, where an exemption for research is in place to ensure the possibility of mining copyrighted works for research purposes [59,60].

Once the datasets were ready, we used automatic language recognition to understand the extent of the cultural diversity of stories available on Wattpad. We preliminarily compared the performances of three different R packages on a sample of around 100,000 titles: the Naïve Bayesian classifier Compact Language Detector v2 (CLD2) [61], the neural network model Compact Language Detector v3 (CLD3) [62], and the n-gram text categorizer TextCat [63]. TextCat was clearly unreliable, whereas CLD2 and CLD3 were not able to identify 24.6% and 29.1% of the titles respectively. Therefore, we decided to use CLD2 for the entire corpus. We then manually checked a random list of 200 titles whose language was not identified and, having determined that the majority of them was in English (78%), we merged them with the English subset in order not to lose 38% of data. We did a second manual check on the 15 most common languages and found that around 100,000 titles were misidentified as Galician, so we merged them with the Spanish titles. In the subset identified as Danish, however, inaccuracy was too high, mixing titles in English, Danish, German and other languages, so we excluded it, since it was only 0.3% of data. As a last step, we decided to use for the analysis of themes only the languages with more than 100,000 titles, that is the first 13 languages of the list.

## 2.2 Stories

Before we started mapping the corpus, we first estimated the number of fanfiction titles by counting terms that refer typically to fanfiction, like "fanfic", "ff", but also names of characters and public figures like "styles" for the singer Harry Styles, "bts" for the Korean band, or

"potter" for Harry Potter, to get a first understanding of the prevalence of this genre in Wattpad. We checked for these terms in the first 1,000 most frequent words in English titles. Then we identified the themes of the stories published analysing the most frequent words for all of the 13 languages with most stories. Stopwords lists for each language have been created starting from those provided by the Stopword ISO repository [64]. We modified the lists in order to remove words specifically related to the writing process ("editing", "completed", "story", etc.) and to keep words that we deemed useful for identifying themes. We did not stop words like "poetry", "poems", "romance", even though they do not convey information about themes, because they can still help in comparing genres popularity between languages. We aggregated records for conjugated lemmas (e.g., the Russian noun "love": "пюбовь", "пюбви"; the Spanish verb "to love": "amo", "amas"), plurals, gendered adjectives and related nouns (e.g., "amigo", "amiga", "amigos", "amistad"). We also manually checked the top 50 words for each language and aggregated synonyms (e.g., Portuguese "girl": "garota", "menina"). For aggregated words we show in the list the one with most occurrences. To interpret the results, we searched the most frequent words directly on Wattpad, in order to understand co-occurrences, clichés, and terms specific of teen fiction self-published stories. For the languages that we do not speak, we used two online dictionaries: translate.google.com and glosbe.com.

## 2.3 Readers

From the subcorpus of 12 stories in English, we extracted and aggregated data about 381,473 users by collecting the number of comments they wrote and which books and paragraphs they commented. A more fine-grained analysis was performed on a sample of 300,000 user profiles. When registering to Wattpad, in fact, users are offered the opportunity to add a short description of themselves and their location. However, this information is optional and fully unstructured, leaving room for creative and unreliable statements. In our sample, only 34.3% of the users provided locations, many of which are fictitious (e.g. "Hogwarts" and "Wonderland"). We converted these locations into standardized state names with the support of the Google Maps Place Autocomplete API [65]. The API produced multiple errors (e.g. "Hogwarts" was located in the United States, "Hell" in Denmark), thus we manually checked and corrected the list. At the end of the process, 61.1% of the locations were correctly georeferenced, while the remaining locations were categorized as fiction (15.8%), house (e.g. "My bed" and "At home", 3.7%), space (e.g. "Pluto" and "Mars", 3.7%), and unidentified (15.6%). Final results were visualized through a map created with Flourish [66].

## 2.4 Engagement

To measure readers' engagement, we used data available on Wattpad but also generated new information by analysing the commenting activity of users extracted from our smallest corpus and doing network analysis.

In a previous study we showed that readers' engagement with TF and Classics is quite different in terms of absolute reads, votes and comments [58]. Here we looked for variations in the engagement with the stories over time, analysing the distribution of reads and comments across paragraphs for Classics and TF. To control whether engagement is affected by personal differences, we also checked whether there are users who read both TF and Classics and behaved differently in their commenting activity in each category. If this is the case, then we can assume that genre has somehow affected the commenting activity. To test this hypothesis, we reduced the scale of our observation, comparing the number of comments by users who read both Classics and TF with the top commentators of Classics. We selected a first sample who wrote at least 200 comments to Classics, and a second sample who wrote at least 100

comments in each of the two categories. We then created Sankey diagrams to represent the flow of comments these users wrote for each book. The aim was to identify the reading preferences of the most active users and to test whether genre is affecting the scale of the commenting activity.

To explore how readers engaged in social interaction with other readers, we generated user-book network graphs for each category. The goal was to visualize the activity of users who read more than one book and whether users did interact with each other across different books. To understand this, we considered both books and users as nodes, aggregating the links between users and paragraphs into user-book links. We limited the text nodes to the level of books not to complicate the network graph excessively. We also generated user-chapter networks graphs for *Pride and Prejudice* and *The Bad Boy's Girl*, which are the two most popular stories in each category. In this case, we considered chapters and users as nodes, aggregating the links between users and paragraphs into user-chapter links. The networks thus generated were still too dense to be analysed (e.g. the full TF network contained about 1.7 million links), thus we reduced their dimensions by cutting less frequent links. The final, simplified networks contained between 1,586 and 3,038 links. The R Markdown file explaining the network analysis preprocessing can be found in the supplementary information S1 File.

Visualizations were realized through the Gephi platform [67]. Among the available layouts, after testing various options, we selected OpenOrd [68] to better highlight distinctions between communities. One of the consequences of its use, is the fact that the weakest links were deleted from the graph, thus visualizations contain only strong connections. Dimensions of the nodes were determined via weighted degree of connection, i.e., by considering both the number of connected nodes and the weight of the links.

The results helped us understand whether readers commented more on the text or replied to other users' comments, and how texts are linked/separated by the commenting activity of users.

## 2.5 Emotions

Further reducing the scale of our investigation, we used quantitative criteria to select excerpts from stories and comments to read closely.

The first criterion used is the number of comments: we identified and read the paragraphs eliciting most comments in each category. The goal was to show what textual elements—story events, style, or else—are more effective in prompting readers' response and what kind of responses they elicit. After that, to understand the link between readers' emotional response and textual features, we used sentiment analysis to search for patterns in the relation between comments and text. Since one of the guiding criteria of our investigation was popularity, we selected the 6 stories that received most reads and votes for each category. Then, for TF, we preprocessed the texts removing paragraphs with paratextual information: mainly author's notes thanking readers, self-promoting, and giving details about story updates. Once the texts have been cleaned, we used the R package Syuzhet [69] to do sentiment analysis [70] and identify positive and negative emotional values in the stories and comments. The purpose of this analysis was to explore the relations between the two emotional arcs, looking for points of convergence and macro-differences between the emotional arc of the story and readers' response.

Syuzhet calculates the sentiment as follows: given a sentiment dictionary that couples a selection of words with negative or positive values, Syuzhet checks if those words are present in the analysed text and sums their values. For longer narrative texts, Syuzhet offers the possibility to visualize the "emotional arcs" of plots by processing the values attributed to each sentence. Such processing works through the application of a series of filters, from the simplest

(moving window) to the most advanced (Fourier transform). Syuzhet has been criticized both for its simplicity and unreliability [71–73]. However, in this application the accuracy of the sentiment analysis with respect to the plot development is not really important, since the focus of our observations is the relations between text's sentiment and comments' sentiment, as a possible signal of remarkable reader response. Moreover, we are working here on a collective scale: the sentiment values are calculated on the sum of all readers' comments. The results express the emotional valence of a collective act of reading, something similar to what Jonathan Rose [11] imagined achieving by studying working-class autobiographies.

We adopted Syuzhet with its default features, by applying the modifications already described in Rebora and Pianzola [34]. Instead of using it to tokenize sentences, we kept the division in paragraphs already provided by Wattpad, in order to have a perfect matching between the text and the associated responses. We then plotted the emotional arcs through the moving window procedure, to keep to a minimum the distortion of the original sentiment values: the scores for each text chunk, in fact, are simply re-calculated as the average of all the surrounding scores (by default, 10% of the total). Finally—and differently from our previous study—we decided not to group all comments in a single text chunk, but to calculate the mean of the sentiment values for each comment. This modification seemed necessary, because Syuzhet does not take into account the frequency of words to calculate sentiments: the scores for longer chunks are statistically more reliable than those for shorter ones, but the impact of words repeated over multiple comments could also get lost, when collapsing them in a single sentence.

The use of Syuzhet for the analysis of Wattpad has also some intrinsic limitations. In the case of Classics, text and comments are from different historical periods, so the use of the same dictionary for sentiment values might cause the misinterpretation of some language uses. For instance, the characters' "decency" and "civility" in talking, typical of Jane Austen's novels, is interpreted as positive, even when the conveyed meaning is negative. In the spirit of literary modelling, comparison could be done using different personalized dictionaries to set the sentiment values of words, adapting them to the language of 19th century English novels or to the specific Wattpad lingo that is found in the comments. Just to refer to a specific case, the word "thanks" occurs in comments to Classics when readers express their gratitude to other users who wrote a comment paraphrasing the paragraph, thus helping other readers who are struggling to understand the text. Syuzhet default dictionary assigns a positive value to "thanks", but in this context it is actually a sign that the text is difficult. If we want to assign consistent values to words expressing the same readers' response, terms like "difficult", "thanks," and "understand" should have similar negative values, because they all represent issues in text comprehension. However, this would create problems in identifying positive uses like, for instance, the hypothetical comment "I understand her feelings" in response to a character's emotions. Moreover, the use of different dictionaries for text and comments would compromise our effort to compare the sentiment values of the two arcs. All considered, we decided to use Syuzhet default lexicon, since it is based on a corpus of novels and it is thus more appropriate for application to literary studies. The R Markdown file explaining the sentiment analysis procedure can be found in the supplementary information S1 File.

Looking for patterns in this relationship between textual features and the emotional response elicited, we first checked for correlation between the two emotional arcs. Then we performed linear regressions to check whether the effect of the story's emotional valence on the readers' emotional response varied between the two genres.

Then, to better understand the patterns spotted and confronting the two genres, we looked at text and comments of the two most popular stories for each category, *Pride and Prejudice* and *The Bad Boy's Girl*. We did an in-depth qualitative analysis of the parts in which we

identified bigger gaps and strongest convergence between the sentiment of story and comments, and analysed the contexts in which the words determining the sentiment values for that segment of text occur. We report here only some examples, but it should be kept in mind that the sentiment value of each segment is determined by all the words to which a value is attributed, sometimes being more than 1,000. For instance, if we want to look at a segment of 10 paragraphs that has an extremely high positive value, we have to consider that Syuzhet, to determine the sentiment, takes words from the segment but also from the 5% of text preceding and following the segment (10% moving window). It may be the case that words in the surrounding text have higher values than words in the selected segment. In this case, for instance, the overall value of the segment may be determined by the combination of 50 positive words with very small values occurring in the selected segment, plus a few words with higher score that occur in the surrounding text. Therefore, interpreting the sentiment values may not always be straightforward for humans, although the sentiment dictionaries have been originally created by human annotators [74,75]. Luckily, this is a kind of effort literary critics are used to: close reading texts paying attention to small details. Once the sentiment outputs are generated and notable segments have been identified, the duty of the critic is to look at the evidence brought by the machine and verify with the text if the argument is solid, even when the machine-generated interpretation is based on a variety of apparently insignificant words. The interpretation of comments' sentiment values is usually more straightforward, since they are often short and a direct expression of readers' thoughts and emotions. However, we already mentioned that the sentiment value for comments is an aggregated datum of the individual values for all comments to one paragraph. Therefore, even with comments it may be the case that a collective negative sentiment is the expression of many negative different words occurring just once, and not the result of a particular story event that elicits clear negative responses, like "I hate this and that".

## 3. Results

### 3.1 Corpus

The highest record number that we identified in the sitemaps was 136,209,997. This is probably the total number of uploads mapped since Wattpad's opening, but we did not find a complete linear progression in the record numbers, since many items may have been deleted by users or not included in the sitemap by Wattpad. The total number of titles we collected is 31,469,668. Wattpad's official statements report that the platform hosts 565 million uploads [76] but we could not verify this number.

Regarding linguistic and cultural diversity, Table 1 shows how many stories have been written for each language and what is their percentage in the corpus. English is by far the most represented language, amounting to around 79% of the Wattpad corpus. However, it should be noted that this is an overestimation: browsing Wattpad, it can be seen that English titles are sometimes also used for stories written in other languages. The extent of the corpus is such that even languages constituting less than 1% of the corpus, like Malaysian, still have more than 100,000 titles.

### 3.2 Stories

Within the English titles subcorpus, there are around 2.3 million fanfiction stories (9.3%), a rough estimation due to the complexity of telling fanfiction apart from other fiction through the analysis of titles only. Thus, the majority of stories published on Wattpad are works of fiction creating original storyworlds. With respect to popular themes, Tables 2–4 show the most frequent words in book titles for the 13 most popular languages on Wattpad.

**Table 1. List of the 13 most used languages on Wattpad, identified with the support of the language detection tool CLD2.**

|  | Language | Number of titles | % of the corpus |
|---|---|---|---|
| 1 | English | 24,869,949 | 79.0% |
| 2 | Spanish | 1,961,109 | 6.2% |
| 3 | Portuguese | 684,004 | 2.2% |
| 4 | Turkish | 560,919 | 1.8% |
| 5 | French | 433,602 | 1.4% |
| 6 | Vietnamese | 431,799 | 1.4% |
| 7 | Indonesian | 304,601 | 1.0% |
| 8 | German | 289,274 | 0.9% |
| 9 | Russian | 264,352 | 0.8% |
| 10 | Italian | 235,665 | 0.7% |
| 11 | Polish | 213,150 | 0.7% |
| 12 | Malaysian | 128,551 | 0.4% |
| 13 | Tagalog (Filipino) | 118,331 | 0.4% |

The complete lists of detected languages can be found in the data repository.

**Table 2. Most frequent words in titles for English, Spanish, Portuguese, Turkish, and French with English translations.**

|  | English | | Spanish | | | Portuguese | | | Turkish | | | French | | |
|---|---|---|---|---|---|---|---|---|---|---|---|---|---|---|
|  | Words | Rel. freq. | Words | Translation | Rel. freq. | Words | Translation | Rel. freq. | Words | Translation | Rel. freq. | Words | Translation | Rel. freq. |
| 1 | love | 171.42 | amor | love | 125.69 | amor | love | 159.45 | aşk | love | 242.04 | amour | love | 168.34 |
| 2 | one | 76.54 | vida | life | 70.62 | Amigo | friend | 143.09 | hayat | life | 70.02 | vie | life | 147.53 |
| 3 | life | 57.10 | amigos | friends | 60.26 | vida | life | 96.71 | kız | girl | 66.10 | coeur | heart | 35.24 |
| 4 | girl | 43.27 | sueño | dream | 46.79 | diário | diary | 86.30 | sözler | words | 46.66 | aimer | to love/like | 34.62 |
| 5 | shot | 33.51 | siempre | forever/always | 42.03 | garota/menina | girl | 73.84 | küçük | small | 40.07 | monde | world | 33.76 |
| 6 | reader | 31.27 | nueva | new | 41.55 | querido | dear/beloved | 47.46 | güzel | beautiful | 39.54 | fille | girl | 33.72 |
| 7 | boy | 27.97 | chica | girl | 32.56 | melhor | best | 45.05 | günlüğü | diary | 38.36 | sans | without | 33.21 |
| 8 | friendship | 25.73 | corazón | heart | 32.20 | dias | days | 35.35 | hayal | dream | 38.20 | amitié | friendship | 30.23 |
| 9 | bad | 23.42 | enamorada | in love | 29.54 | sonhos | dreams | 34.23 | yeni | new | 37.04 | nouvelle | new | 29.74 |
| 10 | heart | 21.27 | día | day | 28.92 | estrellas | stars | 32.64 | karanlık | dark | 30.28 | secret | secret | 28.70 |
| 11 | new | 20.84 | chico | boy | 28.84 | novo | new | 31.09 | yaz | summer | 29.86 | journal | diary | 22.88 |
| 12 | little | 18.70 | sin | without | 24.18 | mundo | world | 27.34 | okul | school | 28.48 | deux | two | 21.64 |
| 13 | lost | 17.26 | nunca | never | 22.72 | nada | nothing | 25.70 | çocuk | child | 26.83 | jour | day | 21.59 |
| 14 | harry | 17.17 | quierer | to want/love | 21.72 | nunca | never | 24.84 | son | last | 26.28 | rencontre | meeting | 20.93 |
| 15 | BTS | 16.62 | mundo | world | 20.42 | irmão | brother | 24.28 | siyah | black | 25.88 | rantbook | rantbook | 19.38 |
| 16 | time | 16.13 | pensamientos | thoughts | 19.36 | sin | without | 23.21 | gerçek | real | 25.56 | jamais | never | 16.00 |
| 17 | best | 14.99 | dos | two | 18.75 | frases | sentences | 22.68 | ilk | first | 24.66 | rêve | dream | 15.19 |
| 18 | never | 14.59 | ojos | eyes | 18.42 | coração | heart | 22.41 | sözleri | remarks | 23.83 | os | oneshot | 15.16 |
| 19 | world | 14.38 | realidad | reality | 16.71 | vez | time | 22.14 | korku | fear/horror | 21.46 | toujours | always | 14.69 |
| 20 | secret | 14.30 | yaoi | Yaoi | 15.54 | filha | daughter | 20.36 | bts | BTS | 21.25 | mort | death | 14.14 |

Words in green cells refer to genres, words in blue cells are fanfiction themes, words in yellow cells are some themes that are unique for that language or unusually high ranking with respect to other languages. The complete lists of words can be found in the data repository. Relative word frequencies are normalised per 10,000 words.

**Table 3. Most frequent words in titles for Vietnamese, Indonesian, German, and Russian with English translations.**

| | Vietnamese | | | Indonesian | | | German | | | Russian | | |
|---|---|---|---|---|---|---|---|---|---|---|---|---|
| | Words | Translation | Rel. freq. | Words | Translation | Rel. freq. | Words | Translation | Rel. freq. | Words | Translation | Rel. freq. |
| 1 | yêu | to love | 199.99 | cinta | love | 207.86 | leben | life | 187.73 | жизнь | love | 285.77 |
| 2 | anh | brother/you | 146.26 | rp | roleplay | 175.02 | Liebe | love | 163.15 | любовь | Life | 282.41 |
| 3 | tình | love | 92.73 | hati | heart | 83.77 | Freundschaft | friendship | 49.34 | дневник | diary | 55.33 |
| 4 | thiên | celestial | 63.35 | memes | memes | 62.26 | neue | new | 38.42 | девушка | girl | 45.68 |
| 5 | sao | star | 57.60 | kata | words | 53.40 | Mädchen | girl | 38.03 | дней | days | 40.84 |
| 6 | sinh | born | 52.35 | teman/persahabatan | friend | 44.65 | Welt | world | 34.45 | друзья | friends | 38.73 |
| 7 | người | person/people | 50.44 | sehun | Sehun (EXO) | 41.20 | immer | always | 28.44 | мир | world | 35.56 |
| 8 | nữ | girl | 49.98 | senja | dusk/evening | 40.91 | Sprüche | claims | 26.13 | один | one | 27.83 |
| 9 | một | one | 46.77 | baekhyun | Baekhyun (EXO) | 38.51 | Tagebuch | diary | 24.48 | просто | simply | 26.95 |
| 10 | đại | great | 43.79 | masa | time/hour | 26.90 | Schicksal | destiny | 21.28 | первая | first | 21.44 |
| 11 | xuyên | to cross | 40.59 | bts | BTS | 24.64 | Geheimnis | secret | 18.04 | без | without | 18.22 |
| 12 | chòm | constellation | 35.57 | saranghae | saranghae | 24.54 | Zeit | time | 17.42 | смерть | death | 17.72 |
| 13 | thần | god | 35.10 | tanpa | without | 23.46 | Tod | death | 17.13 | любить | be in love | 16.56 |
| 14 | oneshot | oneshot | 34.26 | exo | EXO | 23.44 | plötzlich | suddenly | 16.37 | мечты | dreams | 16.35 |
| 15 | tử | death | 33.41 | oppa | oppa | 23.20 | Weg | path | 16.33 | стихи | poems | 16.32 |
| 16 | tiểu | little | 32.89 | diam | silence/secret | 21.99 | verliebt | in love | 15.85 | приключения | adventure | 15.22 |
| 17 | hoa | flower | 32.51 | hujan | rain | 20.54 | Herz | heart | 15.54 | любимый | favourite/beloved | 14.60 |
| 18 | cậu | boy | 29.34 | rindu | miss | 19.28 | zwischen | between | 15.47 | академия | academy | 14.54 |
| 19 | mỹ | danmei | 28.22 | jodoh | mate | 18.88 | zwei | two | 14.53 | подростка | teenager | 14.23 |
| 20 | gia | family | 27.85 | mimpi | dream | 18.79 | Gedanken | thoughts | 14.44 | брат | brother | 14.02 |

Words in green cells refer to genres, words in blue cells are fanfiction themes, words in yellow cells are some themes that are unique for that language or unusually high ranking with respect to other languages. The complete lists of words can be found in the data repository. Relative word frequencies are normalised per 10,000 words.

"Love" is the top word for every language, except German, for which "life" (*leben*) is more frequent. There are words that occur in almost every language, with different proportional frequency: "girl", "life", "friend", "heart", and "dream". There are also some notable exceptions: Indonesian does not mention "girl" and "life" often; Russian, Vietnamese, and Turkish do not mention "heart". Only Western languages mention "world", "new" and "never". "Without" is in the 20 most frequent words for many languages but in English does not occur within the 300 most frequent words. Only a few languages talk about "death" frequently and among them for Polish authors it is quite an important topic. Polish are also the only ones talking about "hope", together with Filipinos. Only in Russian titles "suicides" are quite frequent. There are also unique themes, i.e. words that are quite frequent in only one language. For Tagalog (Filipino): "runaway", which is the third most frequent word, "rich", and a word expressing the surprise for something previously unknown, *pala*. For Malaysian: "memories", "child/son/daughter", and "boss/leader", which can occur both to indicate a superior at work or the leader of a gang. In this second meaning, Malaysian is unique with English in having many stories about "bad boys". Beside Malaysian, family relationships are important for Vietnamese too, as attested by the word "family". Moreover, Vietnamese also has a supernatural positive theme, attested by "celestial", "constellation" and "god". Tagalog (Filipino) is the only language, together with Turkish, to use "beautiful" quite often. Italian and Spanish pay more attention

**Table 4. Most frequent words in titles for Italian, Polish, Malaysian, and Tagalog (Filipino) with English translations.**

| | Italian | | | Polish | | | Malaysian | | | Tagalog (Filipino) | | |
|---|---|---|---|---|---|---|---|---|---|---|---|---|
| | Words | Translation | Rel. freq. | Words | Translation | Rel. freq. | Words | Translation | Rel. freq. | Words | Translation | Rel. freq. |
| 1 | amore | love | 102.59 | miłość | love | 131.08 | cinta | love | 628.25 | mahal | love/affection | 159.78 |
| 2 | ragazza | girl | 63.74 | życie | life | 120.51 | sahabat/kawan | friend | 225.81 | ibig/love | love | 119.91 |
| 3 | vita | life | 62.88 | nominacje | nominations | 95.14 | hati | heart | 97.22 | crush | crush | 113.97 |
| 4 | sogno | dream | 42.14 | przyjaźń | friendship | 42.79 | rasa | feel/think | 69.81 | runaway | runaway | 105.07 |
| 5 | amico | friend | 39.07 | przygody | adventures | 34.03 | hidup | live | 60.26 | shot | shot | 86.85 |
| 6 | occhi | eyes | 38.23 | pamiętnik | diary | 32.26 | sekolah | school | 45.79 | one | one | 78.91 |
| 7 | mondo | world | 35.70 | dziewczyna | girl | 31.27 | diam | silence | 44.19 | buhay | life | 71.18 |
| 8 | sempre | always | 32.56 | shot | shot | 30.85 | takdir | destiny | 39.20 | kaibigan/bestfriend | friend | 64.68 |
| 9 | cuore | heart | 31.25 | nowe | new | 29.14 | senja | dusk/evening | 38.38 | diary | diary | 53.06 |
| 10 | senza | without | 28.57 | zawsze | always | 27.95 | anak | children | 35.11 | boyfriend | boyfriend | 52.15 |
| 11 | citazioni | quotations | 25.52 | świat | world | 26.93 | kenangan | memories | 34.62 | babaeng | girl | 46.74 |
| 12 | ragazzo | boy | 25.19 | śmierć | death | 22.68 | hari | day | 33.95 | sana | hopefully | 33.81 |
| 13 | ultimo | last | 22.07 | tłumaczenie | translation | 22.43 | kata | said | 30.40 | hugot | lines | 33.18 |
| 14 | mai | never | 21.86 | kocham | to love | 17.86 | kelas | class | 30.27 | tula | poem | 32.26 |
| 15 | diario | diary | 21.80 | cytaty | quotes | 17.27 | rumah | home | 30.06 | talaga | really | 30.08 |
| 16 | migliore | best | 20.28 | marzenia | dreams | 14.43 | ketua | chief/leader | 29.91 | pala | "revelation"/"surprise" | 28.98 |
| 17 | innamorata | in love | 20.02 | dzień | day | 14.26 | bersama | together | 29.19 | kaya | rich | 28.47 |
| 18 | amo | to love | 19.55 | nigdy | never | 14.22 | bahagia | happy | 28.73 | puso | heart | 27.61 |
| 19 | inizio | beginning | 17.05 | szkoła | school | 13.62 | sekai | Sehun/Kai (EXO) | 28.11 | rin | well | 26.86 |
| 20 | fine | end | 14.13 | początek | beginning | 13.24 | pertama | first | 28.08 | pangarap | dream | 25.81 |

Words in green cells refer to genres, words in blue cells are fanfiction themes, words in yellow cells are some themes that are unique for that language or unusually high ranking with respect to other languages. The complete lists of words can be found in the data repository. Relative word frequencies are normalised per 10,000 words.

than other languages to physical details, especially the "eyes". Regarding female roles: "princess" is frequent only for English, Polish, and Russian.

In many languages, there are two groups of words that give us more information about reading preferences, those about fanfiction and those about literary genres. Inspiration for fanfiction is different worldwide, in some languages is dominant while in others does not occur at all in the 50 most frequent words. References that are extremely popular in English, like Harry Potter and Harry Styles (a singer), do not occur in other languages. The only fanfiction theme which is common across more languages is K-pop, with stories about the two most famous boy bands, "BTS" and "EXO". In Indonesian, Korean words like *saranghae* and *oppa* may refer to fanfiction but have also been absorbed in language uses outside the genre, given the influence of Korean culture in South East Asian countries. The analysis of words also revealed preferences for genres. Collections of very brief compositions are quite popular in many languages ("thoughts", "lines", "sentences", "quotes", "words", "remarks"). "Diary" (including "rantbook") and "oneshot", too, have a transnational success. The latter are stories consisting of only one chapter, whose length can vary from a few hundred words up to a few thousands. English shows a peculiar preference for the "x reader" genre, which refers to stories written in second or first person and whose protagonist is the reader [77]. In Malaysian, "roleplay" and

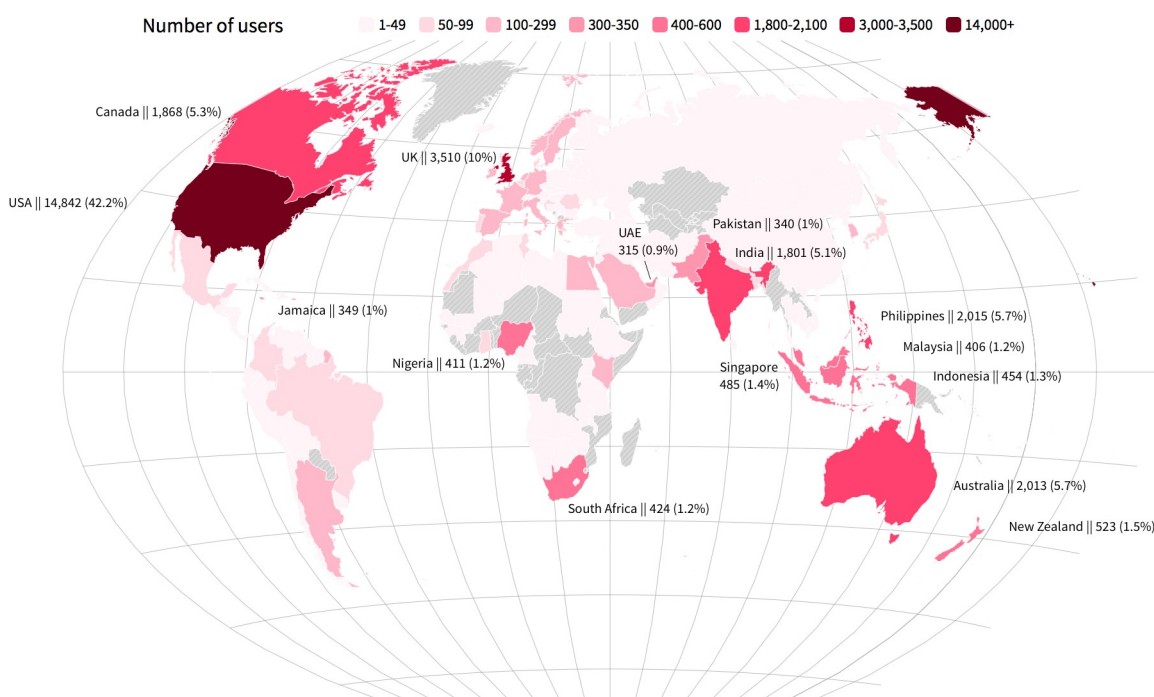

**Fig 1. Map of users' location extracted from a corpus of 12 English language stories (*n* = 35,208).** Only details for countries with more than 300 users are written. An interactive version of the map with data for all countries is available as supporting information (S1 Fig). Reprinted from http://flourish.studio under a CC BY license, with permission from Flourish, Inc., original copyright 2019.

"memes" are clearly the dominating genres. Turkish have a preference for "horror" stories. Vietnamese and Spanish readers like "danmei" and "yaoi", that is stories about male homosexual relationship targeted to female audience [78].

## 3.3 Readers

Some statistics about Wattpad readers can be obtained from the website itself: more than 70 million people use it every month, 90% of whom are "either Generation Z or Millennial", i.e. younger than 35 years old [79]. More precisely, according to a survey conducted by Wattpad with 650 English speaking users worldwide, 80% of the respondents are female between 13 and 24 years old [80]. The average reading session is 37 minutes long and 85% of the times from a mobile phone [79,81]. The survey also states that "Almost half of respondents bought one or two books a month", widely preferring paperbacks to ebooks, because readers "see themselves as curators and love to feel the paper, annotate their books, and financially support authors". Beside the fact that paper books have the advantage of not running out of battery [80].

The sample of respondents to Wattpad's survey is similar to the one we extracted from our comments corpus based on 12 titles from TF and Classics, in terms of language spoken (English) and location (worldwide). The sample we created includes 300,000 users, with real geographical location information available only for 35,208 of them. Fig 1 is a map of users' distribution, showing that stories written in English are read by users living in many different countries. Most readers are from countries where English is either the first language (USA, UK, Canada, etc.) or a second language (India, Philippines, Indonesia, etc.), but there are also many non-English speaking countries, each of them amounting to around 0.5% of the readership (e.g., almost all European countries, Saudi Arabia, Egypt, South Korea, Argentina, etc.). Among the 12 stories in our corpus, Classic authors are all from the UK and TF authors come

from the USA, Canada, and one from South Africa, but the works by all of them have been read in all continents. There is a worldwide community of readers who read, comment, and discuss books over the years, engaging with the reflections and opinions of people living in various countries, and meeting other culture-specific responses to the same stories.

### 3.4 Engagement

Rebora and Pianzola [34] have shown that the number of reads, votes, and comments for TF and Classics is quite different, reaching a maximum of 197 million reads and 2.5 million comments for TF, but only 7.4 million reads and 42,000 comments for Classics (Table 5).

Absolute values can only suggest a first idea of readers' engagement, similar to knowing how many people bought a book, without actually knowing whether they read it or just skimmed the first pages to see whether they may like it. Indeed, even among the commentators, there is a substantial amount of readers who only wrote one comment, usually to the book title or the first paragraph. Figs 2 and 3 report the distribution of number of reads and comments across paragraphs for 20 TF and Classics titles, an information that can help us better understand how many readers actually read the whole books, commenting while they progressed in reading. It is evident that very often users begin to read Classics but abandon them after the initial chapters. This phenomenon is mirrored by the commenting activity, with the

**Table 5. Statistics for the Classics and Teen Fiction categories on wattpad.**

| | Classics | | | | | Teen Fiction | | | |
|---|---|---|---|---|---|---|---|---|---|
| | Book | Total comments | Read count | Vote count | | Book | Total comments | Read count | Vote count |
| 1 | *Pride and Prejudice* | 42,013 | 7,400,000 | 113,000 | 1 | *The Bad Boy's Girl* | 2,569,405 | 197,000,000 | 3,400,000 |
| 2 | *Romeo and Juliet* | 11,607 | 3,100,000 | 36,700 | 2 | *I Sold Myself to the Devil for Vinyls . . . Pitiful I Know* | 2,052,682 | 92,900,000 | 2,000,000 |
| 3 | *Wuthering Heights* | 6,653 | 1,700,000 | 13,200 | 3 | *She's With Me* | 1,788,844 | 102,000,000 | 3,700,000 |
| 4 | *Jane Eyre* | 6,177 | 1,600,000 | 16,700 | 4 | *The Hoodie Girl* | 1,567,444 | 58,000,000 | 2,200,000 |
| 5 | *Alice's Adventures in Wonderland* | 3,261 | 1,100,000 | 11,100 | 5 | *The Last Virgin Standing* | 1,412,758 | 61,900,000 | 1,600,000 |
| 6 | *The Picture of Dorian Gray* | 2,768 | 1,000,000 | 8,800 | 6 | *My Brother's Best Friend* | 1,204,380 | 114,000,000 | 2,200,000 |
| 7 | *Emma* | 2,137 | 1,200,000 | 8,900 | 7 | *The Cell Phone Swap* | 1,118,017 | 100,000,000 | 2,100,000 |
| 8 | *Great Expectations* | 1,767 | 1,300,000 | 8,500 | 8 | *The Bad Boy, Cupid & Me* | 1,004,800 | 64,000,000 | 1,700,000 |
| 9 | *Little Women* | 1,636 | 498,000 | 9,300 | 9 | *Mr. Popular and I* | 843,820 | 99,000,000 | 1,700,000 |
| 10 | *Anna Karenina* | 1,595 | 1,100,000 | 16,700 | 10 | *My Wattpad Love* | 733,900 | 47,200,000 | 1,400,000 |
| 11 | *Dracula* | 1,546 | 290,000 | 4,800 | 11 | *Breaking the Bad Boy* | 721,200 | 29,100,000 | 978,000 |
| 12 | *Anne of Green Gables* | 1,255 | 389,000 | 9,800 | 12 | *Stay With Me* | 682,194 | 25,800,000 | 1,200,000 |
| 13 | *The Adventures of Sherlock Holmes* | 1,232 | 454,000 | 6,500 | 13 | *Bad Boy's Game* | 668,489 | 52,000,000 | 1,600,000 |
| 14 | *A Tale of Two Cities* | 1,034 | 300,000 | 3,400 | 14 | *Must Date The PLAYBOY!* | 661,865 | 100,000,000 | 1,700,000 |
| 15 | *Macbeth* | 1,021 | 125,000 | 1,900 | 15 | *Growing up (MWL's sequel)* | 659,900 | 23,500,000 | 760,000 |
| 16 | *The Importance of Being Earnest* | 975 | 134,000 | 1,800 | 16 | *The Quirky Tale of April Hale (Quirky Series #1)* | 637,304 | 43,200,000 | 1,100,000 |
| 17 | *A Midsummer Night's Dream* | 845 | 112,000 | 2,000 | 17 | *Silently Falling* | 608,528 | 24,300,000 | 1,100,000 |
| 18 | *Demian* | 769 | 79,600 | 1,400 | 18 | *The President's Daughter* | 569,000 | 42,900,000 | 1,100,000 |
| 19 | *Hamlet* | 757 | 140,000 | 2,000 | 19 | *Started With a Lie* | 554,976 | 49,800,000 | 1,000,000 |
| 20 | *Oliver Twist* | 719 | 280,000 | 4,100 | 20 | *Just A Friend?* | 554,208 | 36,300,000 | 1,000,000 |

Originally published in Rebora and Pianzola [34].

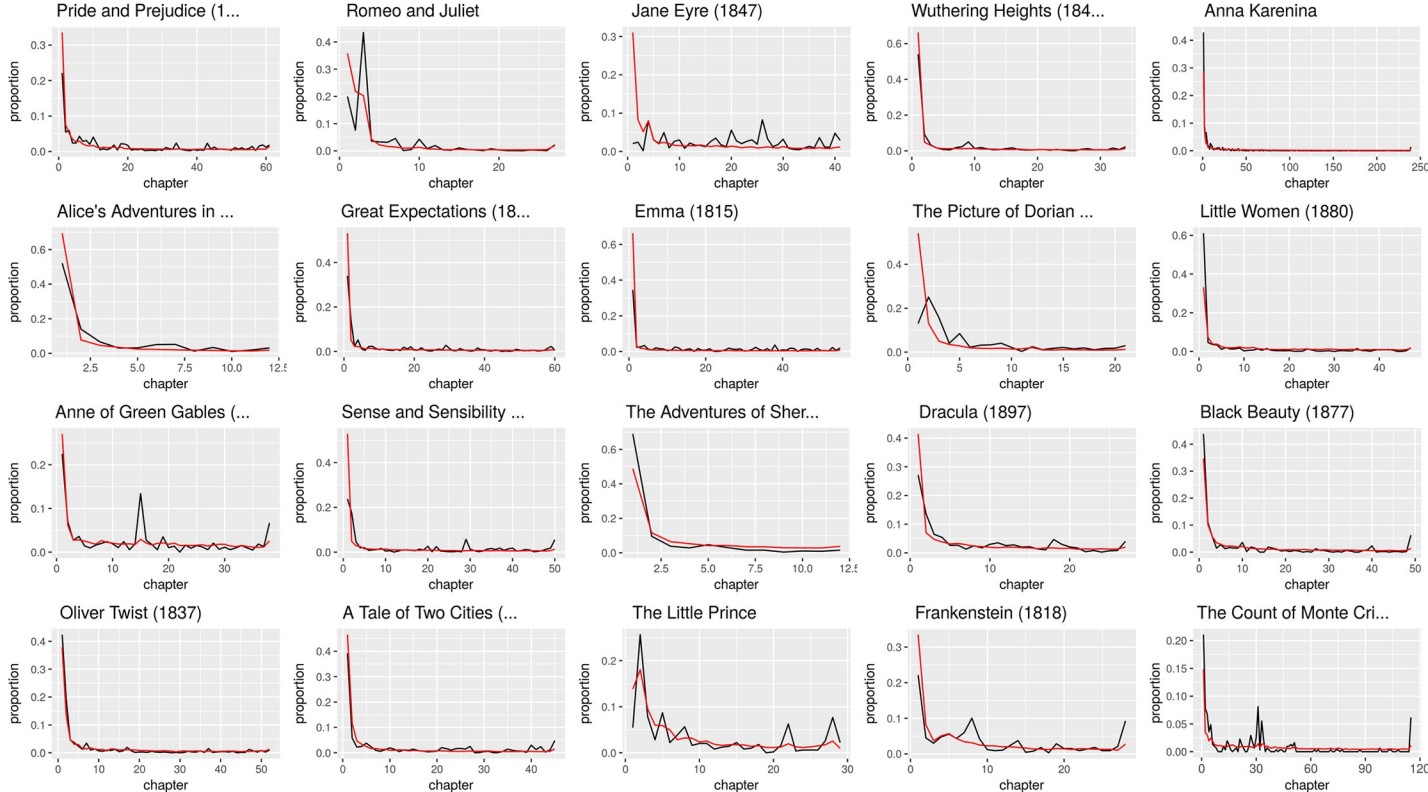

**Fig 2. Distribution of number of reads and comments across chapters for 20 Classic stories.** The red line indicates the reads, the black line indicates the comments. The scale of the Y axis is the proportion of the total of reads and comments per title, to make the plots comparable: all chapters' values have been divided by the total sum of reads or comments in the book.

only exception of *Jane Eyre*. On the other hand, TF stories are able to engage readers for a long part of the plot development, in some cases even increasing the number of reads and comments in later chapters.

The next step was to look for quantitative variations in the commenting activity of the most active users. Among all users, 19 wrote at least 200 comments to Classics and 18 wrote at least 100 comments in both categories. The two Sankey diagrams below (Figs 4 and 5) show the different behaviour of these two samples. Users are in the centre with flow lines going left and right to book titles: TF on the left, Classics on the right. The width of the lines connecting users and books is proportional to the number of comments written on that story by the user. The numbers on the links indicate how many comments a user has written to that specific story. All data and interactive versions of the figures, in which all link values can be seen, are available as supporting information.

The results of our analyses of the flow of comments show quite different results for the samples of commentators of both categories and for the sample of commentators of Classics. Fig 4 shows the flow of comments for the 19 readers who wrote at least 200 comments to Classics. There are only 3 users who also wrote at least 100 comments to TF. The attention of strong commentators of Classics is quite polarized: 84% of them did not read TF. Among the Classics, *Pride and Prejudice*, *Jane Eyre*, and *Wuthering Heights* were favourite reads, whereas *Emma* and *Alice's Adventures in Wonderland* had only one very devoted commentator each (users FF and JJ, respectively).

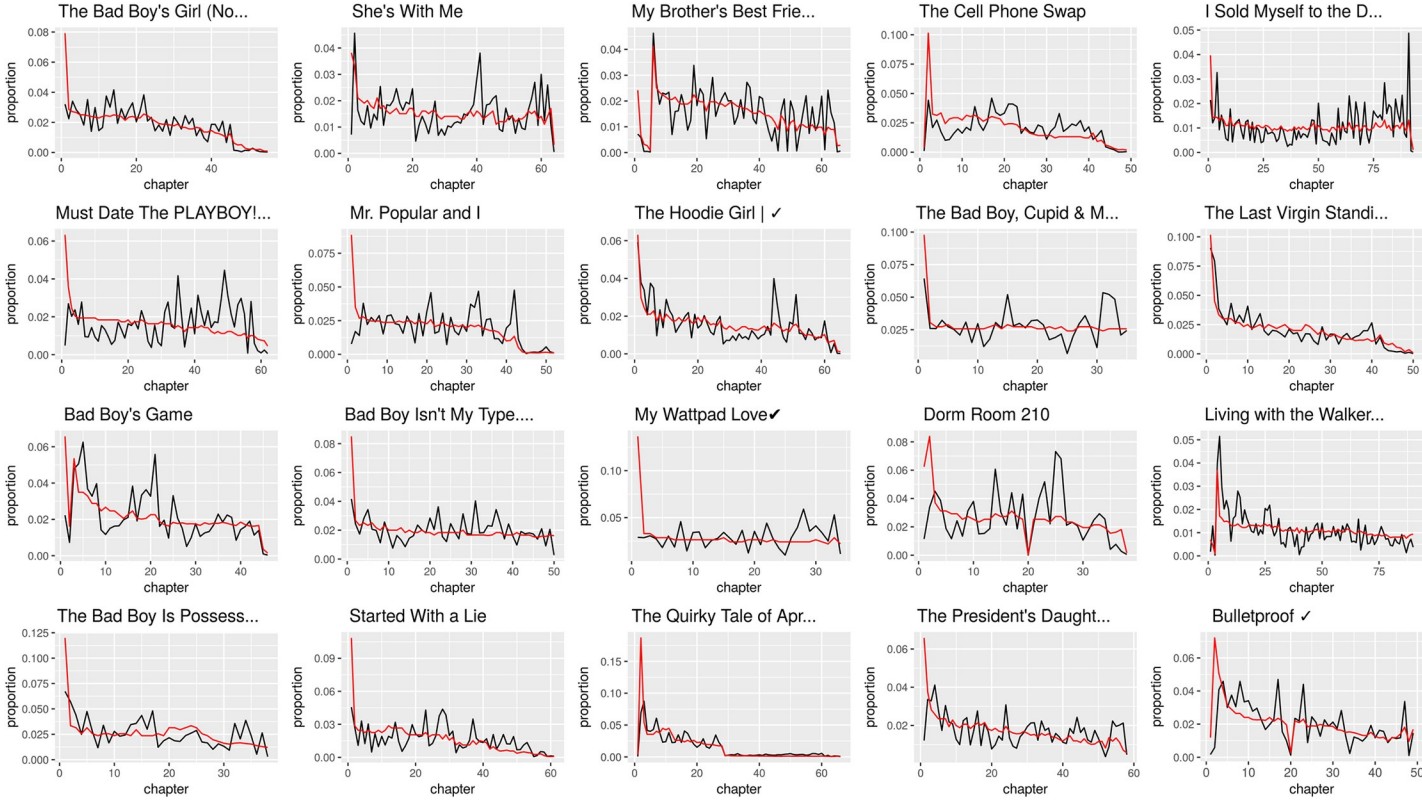

**Fig 3. Distribution of number of reads and comments across chapters for 20 Teen Fiction stories.** The red line indicates the reads, the black line indicates the comments. The scale of the Y axis is the proportion of the total of reads and comments per title, to make the plots comparable: all chapters' values have been divided by the total sum of reads or comments in the book.

Fig 5 shows which books, and how much, have been commented by users that were very active in reading both Classics and TF socially. Among them, only 5 readers out of 18 wrote more comments to Classics (users Q, G, H, R, P), but without showing a dominant preference for this genre. They focused mainly on three titles—*Pride and Prejudice*, *Jane Eyre*, and *Wuthering Heights*—only user G also commented on *Emma* 5 times. Overall, the majority of the 18 readers read and commented on TF more.

Beside the reading interests of top users, we also wanted to get a sense of the reads shared by a broader sample of users and to understand how important social interactions are when reading. The two users-books networks below (Figs 6–8) visualise the relative vicinity of books —based on the readers they have in common—and the level of interaction between users across different books and within each genre. Yellow and blue nodes are books (yellow for Classics, blue for TF), pink nodes are users. The size of the nodes is proportional to the number of connections the node has. The colour of the edges is yellow/blue to connect users to the books they commented and pink to connect users that replied to each other's comments. The width of the edges is proportional to the number of interactions. Before interpreting the results, it is worth reporting some general statistics for the two categories: for TF, just 21.7% of the comments are answers to other commentators, while the percentage increases to 30.6% for the Classics.

The users-Classics network (Fig 6) shows the isolation of *Emma* and *Alice's Adventures in Wonderland*: they do not share many readers with other stories and there are no strong interactions between their readers. There is a central conglomerate with *Pride and Prejudice*, *Jane*

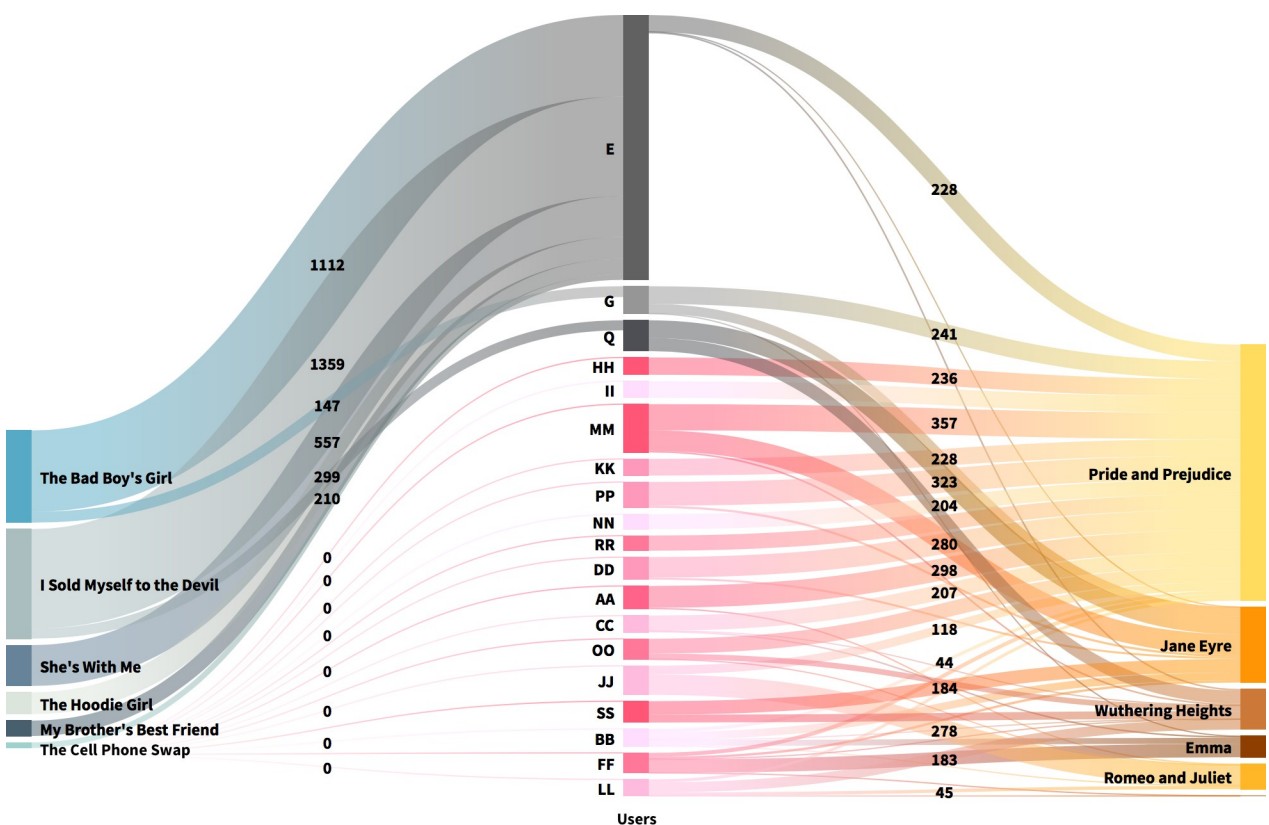

**Fig 4. Flow of comments for readers who wrote at least 200 comments to Classics.** Users are in the centre with flow lines going left and right to book titles; TF on the left, Classics on the right. Users represented with grey tones also wrote at least 100 comments to Teen Fiction. The width of the lines connecting users and books is proportional to the number of comments written on that story by the user. The number on the links indicate the number of comments written by a user to that specific story. An interactive version of this figure, in which all values can be seen, is available as supporting information (S2 Fig). Reprinted from http://flourish.studio under a CC BY license, with permission from Flourish, Inc., original copyright 2019.

*Eyre*, and *Wuthering Heights*, which share many readers with a remarkable amount of interactions (Fig 7; see the pink edges connecting the pink nodes). For instance, the node USER_394 has a weighted degree of 11.73 (corresponding to just 2% of the maximum, 547.97 for *Pride and Prejudice*) and distances itself from *Pride and Prejudice*, notwithstanding the strong connection it has with this title. Its position depends mainly on the wide range of relationships it establishes with readers of other novels (the thinner pink edges). However, it should be noted that the strongest user in this graph (USER_80, with a weighted degree of 19.02) is represented by the node placing itself immediately above *Pride and Prejudice*, despite having strong connections with *Jane Eyre* and *Wuthering Heights*, too.

The isolated clusters of a few pink nodes indicate that there are some readers who have stronger connections with other readers than with the text. For them the conversational aspect of social reading is more important than commenting on the text, as suggested by their distance from the yellow book nodes. The reason for this became clear reading the comments: since Classics are written in an English difficult to understand for many teenagers, and depict worlds with social norms not familiar to them, conversations in the comments are often requests for help in understanding what is happening in the story.

In the TF graph (Fig 8), it can be noted that a title like *I Sold Myself to the Devil* (Eigenvector centrality = 1), even if less commented overall, is a stronger attractor than *The Bad Boy's Girl*

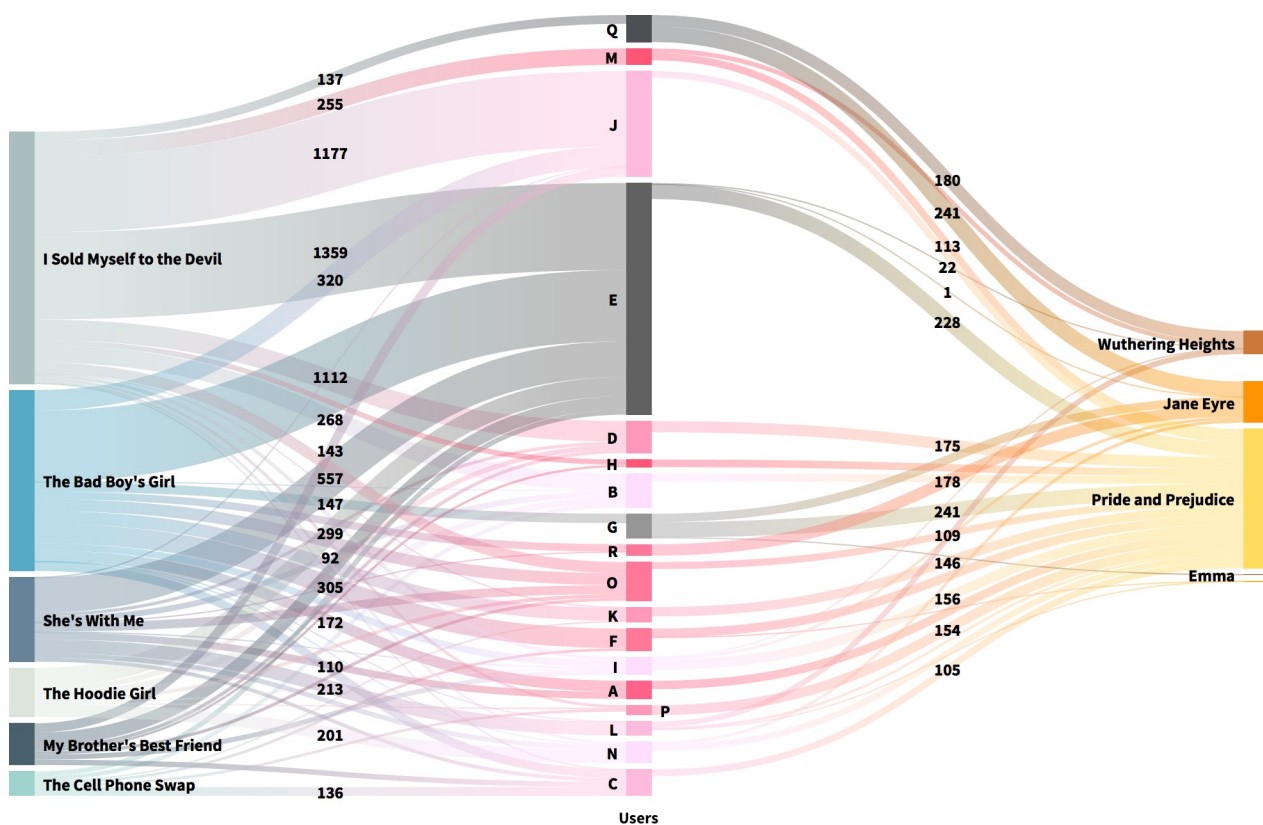

**Fig 5. Flow of comments for readers who wrote at least 100 comments to both Classics and TF.** Users (*n* = 18) are in the centre with flow lines going left and right to book titles: TF on the left, Classics on the right. Users represented with grey tones also wrote at least 200 comments to Classics. The width of the lines connecting users and books is proportional to the number of comments written on that story by the user. The number on the links indicate the number of comments written by a user to that specific story. An interactive version of this figure, in which all values can be seen, is available as supporting information (S3 Fig). Reprinted from http://flourish.studio under a CC BY license, with permission from Flourish, Inc., original copyright 2019.

(Eigenvector centrality = 0.90), because it has more very active commentators than other titles. The pink clusters at the margins of the graph are very sociable users who engaged repeatedly with 1–5 other readers more than 250 times. One peculiar exception to this trend is represented by the fact that one of the strongest user nodes (USER_2411, with a weighted degree of 23.99) places itself at the border of the network, sustained almost uniquely by relations with other users (the only connection with a book is directed towards *I Sold Myself to the Devil*). The relevance of this node is only slightly surpassed by that of USER_358, that connects itself with almost all the TF titles (and with no other user). Note that the node places itself very close to *The Cell Phone Swap* but has no connections with it. The position is determined by the relations it has with the other five titles.

After having analysed the connections within the two genres, we then looked at how users commented on two stories—*Pride and Prejudice* and *The Bad Boy's Girl*—and engaged in social reading. Figs 9–12 show the users-chapters network graphs for each title. In the *Pride and Prejudice* network graph (Figs 9 and 10), the first chapters are isolated and clearly surrounded by many users who do not have connections to other chapters. This means that they did not continue reading past the beginning of the story, as it is also confirmed by the higher number of comments to these chapters in comparison to the others (see comments distribution in Fig 3 above). The graph confirms the high user-user activity seen in the Classics

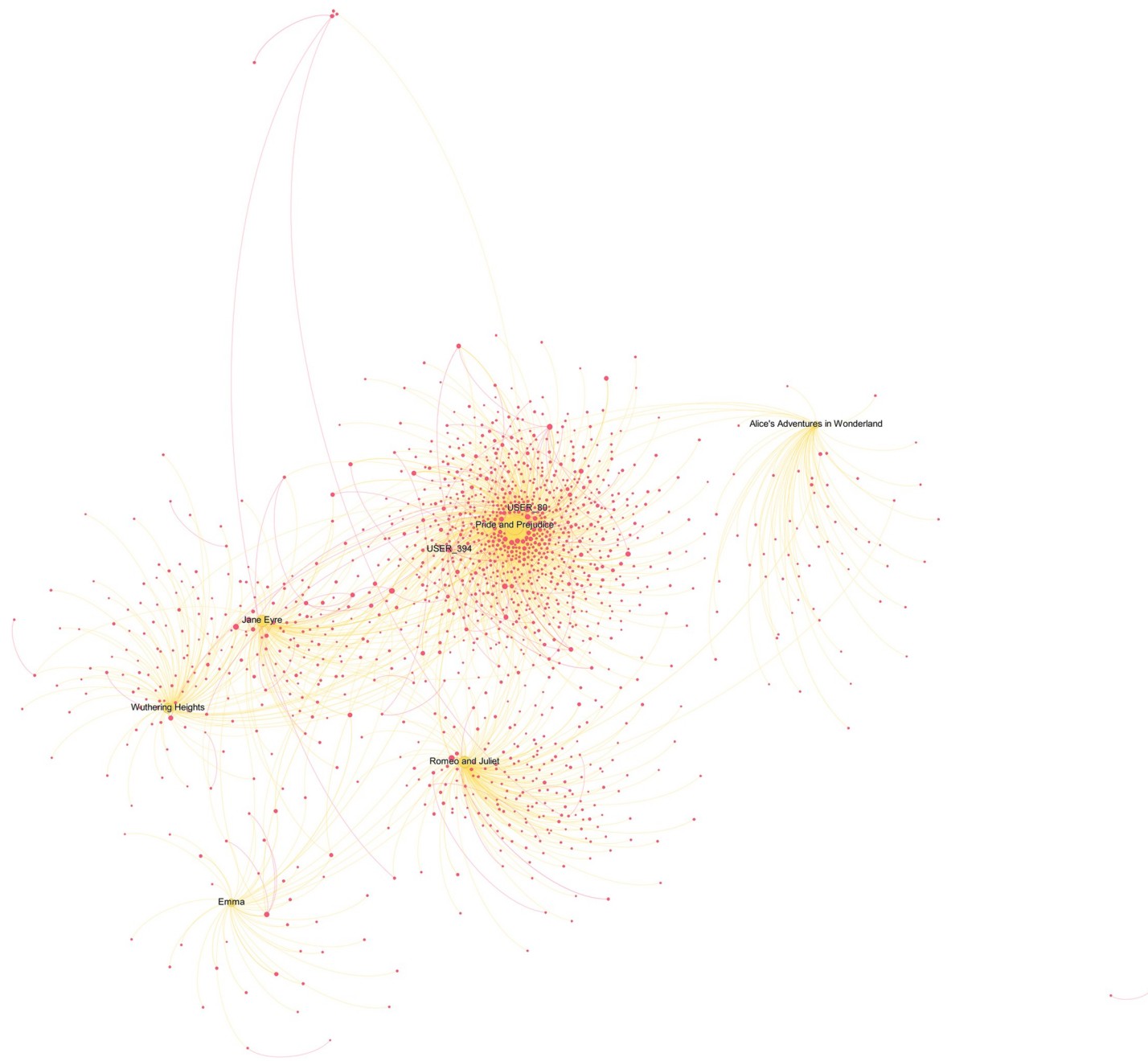

**Fig 6. Classics users-books network graph (detail 1).** The graph contains 1,433 nodes, 12.4% of the total in the category (11,583). This reduction was obtained by taking into consideration only nodes that interacted between each other at least 5 times. Weighted degree of the nodes (i.e. their dimension) is visualized through a logarithmic scale, thus half diameter corresponds to 1/10 of weighted degree. We chose this option to better highlight distinctions. Due to the large dimension of the network, some nodes remained out of the figure. A full-scale figure, as well as an interactive Gephi file are available as supporting information (S4 Fig and S2 File).

network (Figs 6 and 7): readers talk between each other more than in *The Bad Boy's Girl* (Figs 11 and 12). Reading the comments, we found that, in the case of *Pride and Prejudice*, the comments to the first chapters are often questions like "Is this a complete copy?" or "i dont understand . . . is this the real book?", but also "Is this for real? Jane Austen has a wattpad account . . . Do you have a copy right or whatever? Just asking..". These are paratextual questions likely to

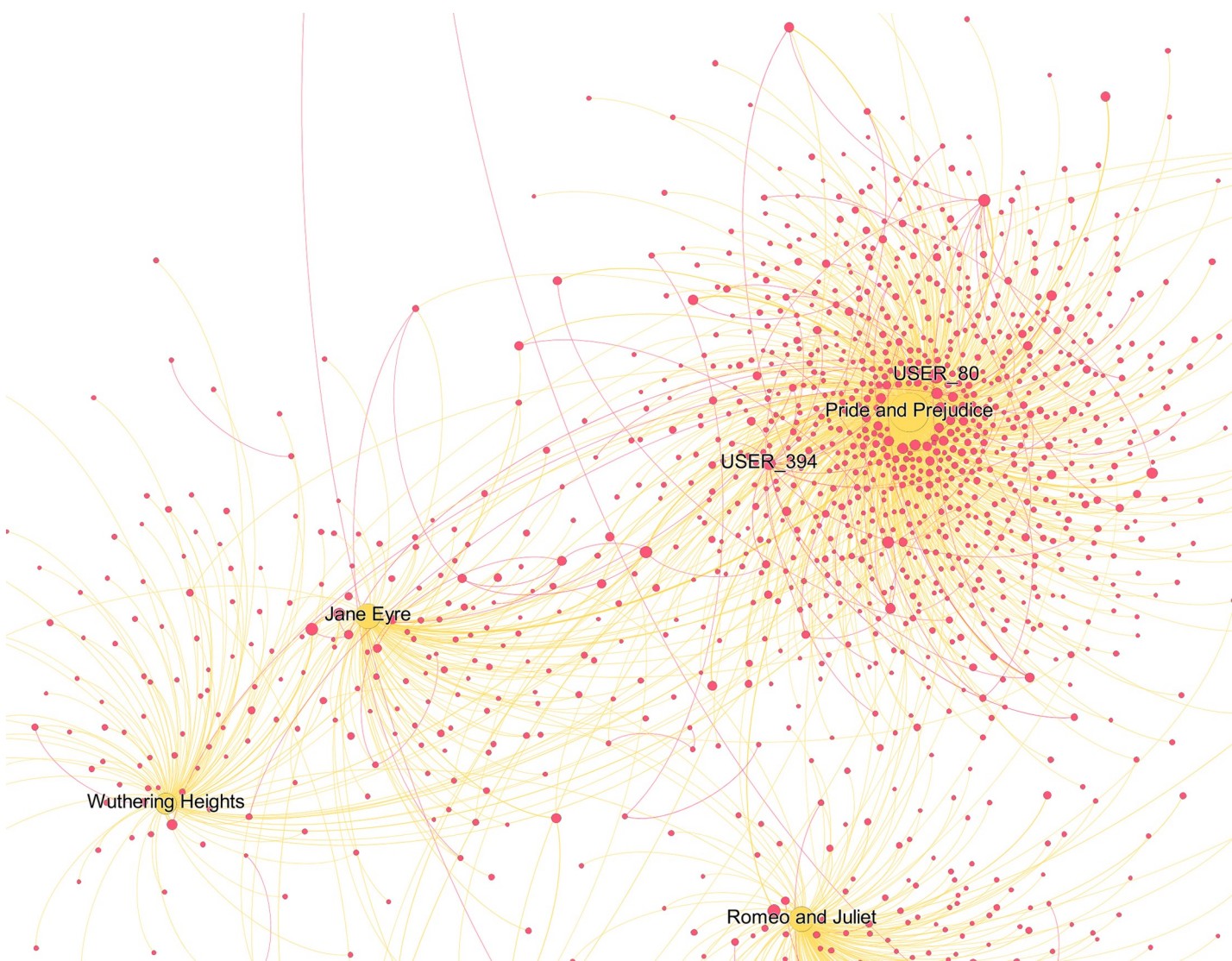

**Fig 7. Classics users-books network graph (detail 2).** The full graph contains 1,433 nodes, 12.4% of the total in the category (11,583). This reduction was obtained by taking into consideration only nodes that interacted between each other at least 5 times. Weighted degree of the nodes (i.e. their dimension) is visualized through a logarithmic scale, thus half diameter corresponds to 1/10 of the weighted degree. We chose this option to better highlight distinctions.

start conversations between users before they actually begin to engage with the story. There are two major clusters: on the right, until chapter 15; on the left, chapters from 16 to 61. From chapter 16 the group of commentators is more stable: the left area is dense with user-chapter connections (yellow edges) and there are more big user nodes compared to the first 14 chapters. This means that the activity of individual users intensified after the first quarter of the story. The right cluster is stretched because of the attraction of a very sociable user (USER_277, with a weighted degree of 87.27), who had many social interactions from the beginning of the book and pulls the connections of the clusters towards the top.

*The Bad Boy's Girl* graph (Fig 11) confirms two trends seen in the whole TF category (Fig 8), namely that there are many very active commentators, as attested by the size of the user nodes, but also a few single users who form separate clusters of readers. The biggest of these nodes (USER_682) even reaches a weighted degree (63.69) comparable to that of the strongest

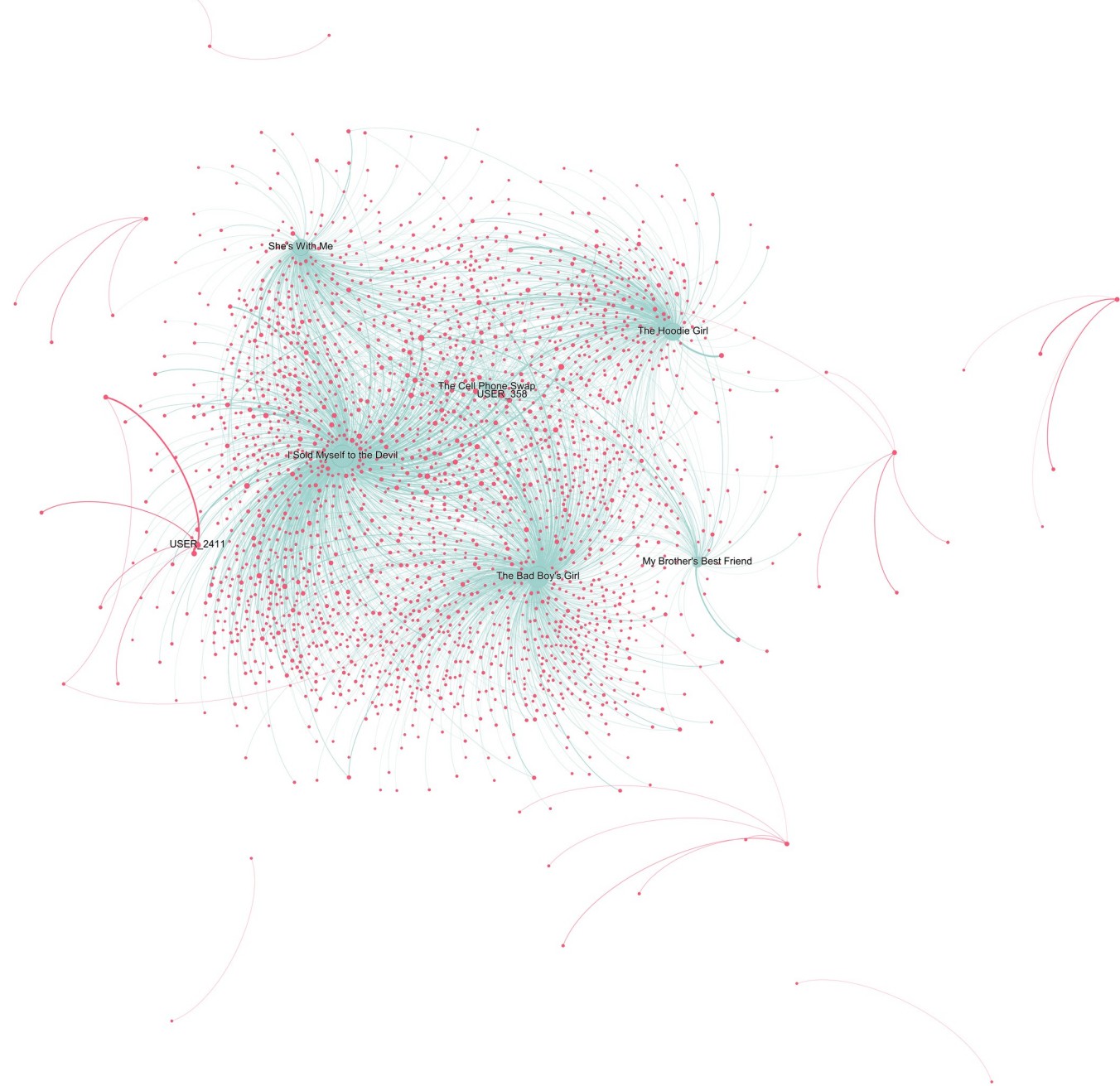

**Fig 8. Teen Fiction users-books network graph.** The graph contains 2,423 nodes, 0.7% of the total in the category (347,884). This reduction was obtained by taking into consideration only nodes that interacted between each other at least 250 times. Weighted degree of the nodes (i.e. their dimension) is visualized through a logarithmic scale, thus half diameter corresponds to 1/10 of the weighted degree. We chose this option to better highlight distinctions. Due to the large dimension of the network, the quality of the image is limited. A full-scale figure, as well as an interactive Gephi file are available as supporting information (S5 Fig and S3 File).

chapter node (CHAP_6, 119.93) (Fig 12). Users clusters are so relevant in the network that they can even attract chapters isolating them from the main cluster, as it is the case for Chapter 5. Its position is not due to low engagement (it ranks 32$^{nd}$ out of 42, with 46,700 comments vs. 16,900 of the least commented chapters; see dataset), but rather to the attraction of a user who commented it a lot and had another user as a privileged discussant (Fig 12).

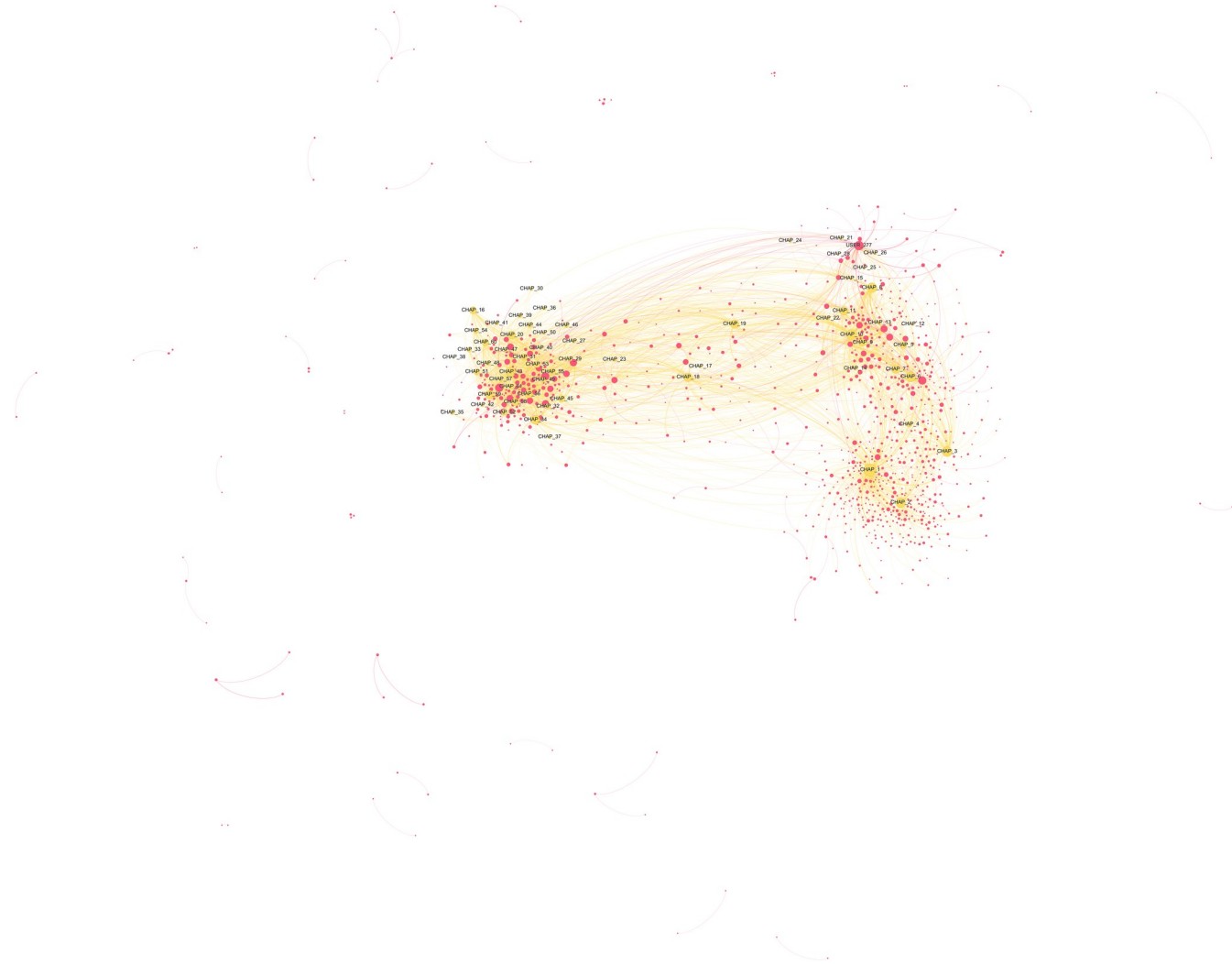

**Fig 9. Users-chapters network graph for *Pride and Prejudice*.** The full graph contains 1,067 nodes, 17% of the total in the book (6,278). This reduction was obtained by taking into consideration only nodes that interacted between each other at least 3 times. Weighted degree of the nodes (i.e. their dimension) is visualized through a logarithmic scale, thus half diameter corresponds to 1/10 of the weighted degree. We chose this option to better highlight distinctions. Due to the large dimension of the network, the quality of the image is limited. A high-resolution figure, as well as an interactive Gephi file are available as supporting information (S6 Fig and S4 File).

### 3.5 Emotions

To understand what text features elicit the strongest emotional response, we began by identifying the paragraphs that generate more interest, in terms of number of comments. Table 6 shows the most commented paragraph in the categories Classics and TF.

Within Classics, as expected, the most commented paragraphs belong to *Pride and Prejudice*, since there is a big difference in the total number of readers for this book and the others of the category. Previously, Fig 2 showed that for Classics the first paragraphs are usually the most commented of the book. This is because Wattpad readers like to announce publicly that they started reading a story, often stating whether it is their first time reading it ("ftr", first time reader) or they are rereading it ("rr", rereading). Browsing through the comments to the incipits of *Pride and Prejudice* and *Wuthering Heights*, it became clear that the success of these two books is due to the fact that they are mentioned in the TF novel *After* [82], in which two

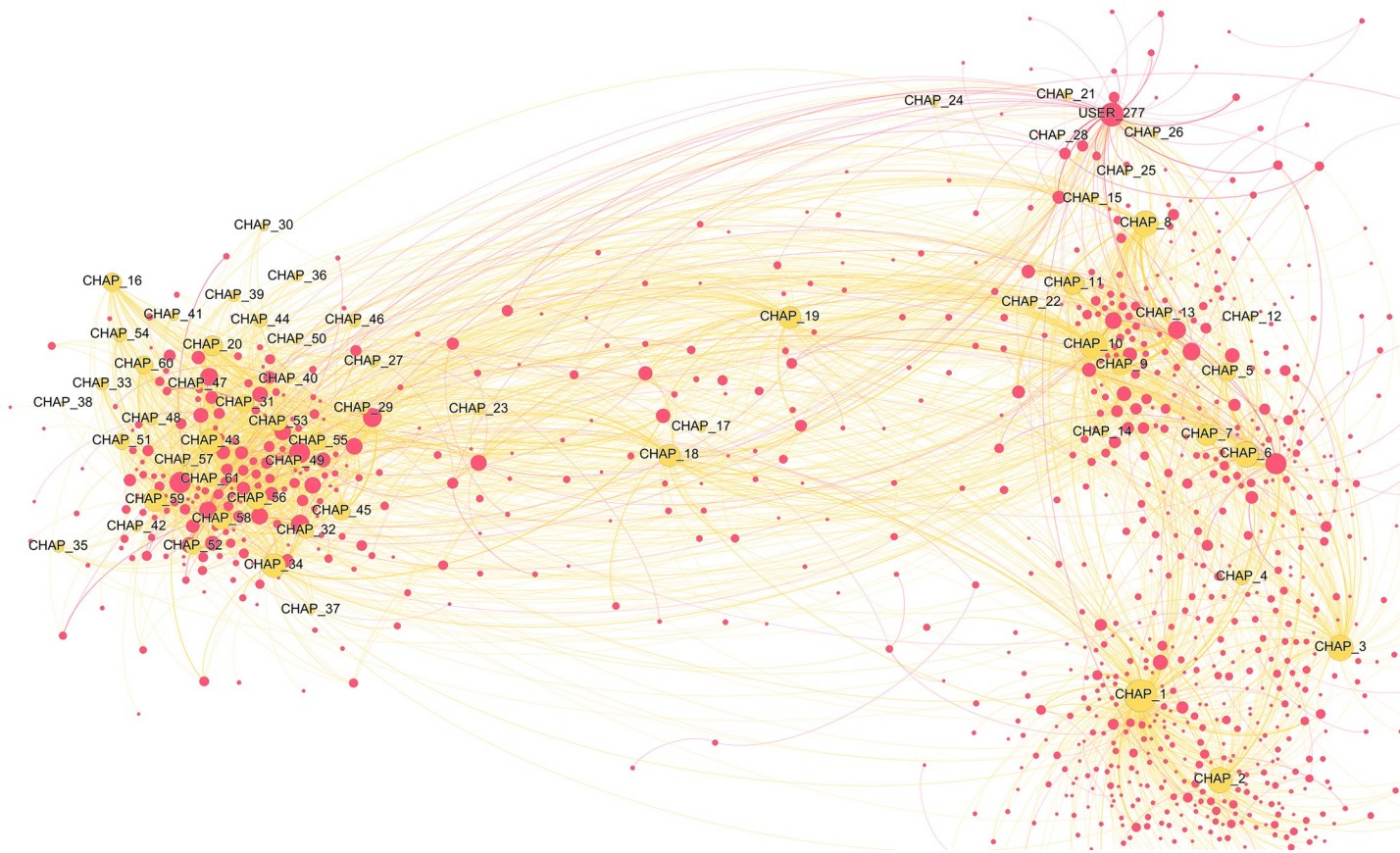

**Fig 10. Users-chapters network graph for *Pride and Prejudice* (detail).** The full graph contains 1,067 nodes, 17% of the total in the book (6,278). This reduction was obtained by taking into consideration only nodes that interacted between each other at least 3 times. Weighted degree of the nodes (i.e. their dimension) is visualized through a logarithmic scale, thus half diameter corresponds to 1/10 of the weighted degree. We chose this option to better highlight distinctions.

characters discuss Emily Brontë's book and, in another scene, one male character says: "Elizabeth Bennet needs to chill". Many readers of *After* claim that they have been attracted to the two Classics for this reason. There are also many other readers that use Wattpad for their English class required readings. Something similar happens for *Romeo and Juliet*: many readers claim to either read it for school or because of the song "The Prologue" by the American singer Halsey, which begins with Shakespeare's prologue. The second most commented paragraph of the tragedy, however, is a first example of how content, rather than external factors, can spark readers' reaction. It is the line by Sampson—of the house of Capulet—who says that he will rape Montague women. Readers react to this claim insulting the character and explaining to other readers the meaning of the allusion "thrust his maids to the wall", since 17th century English may not be easily understood by everybody.

The two other highly commented paragraphs in the first chapter of *Pride and Prejudice* are examples of a trend that later continues in the whole book. First, Readers dislike Mrs. Bennet—especially the way she talks about Elizabeth. They react directly to her behaviour by saying she is annoying, or rude, or by criticising her role as a mother, etc. Readers also elaborate more on how her opinions denounce aspects of 19th century British society, like "Back in the 1800's people prefer beauty over brains. A concept I can never agree with". The second trend concerns Mr. Bennet's wit and sarcasm. Readers love the way he often replies to his wife. For instance, for the paragraph "You mistake me, my dear. I have a high respect for your

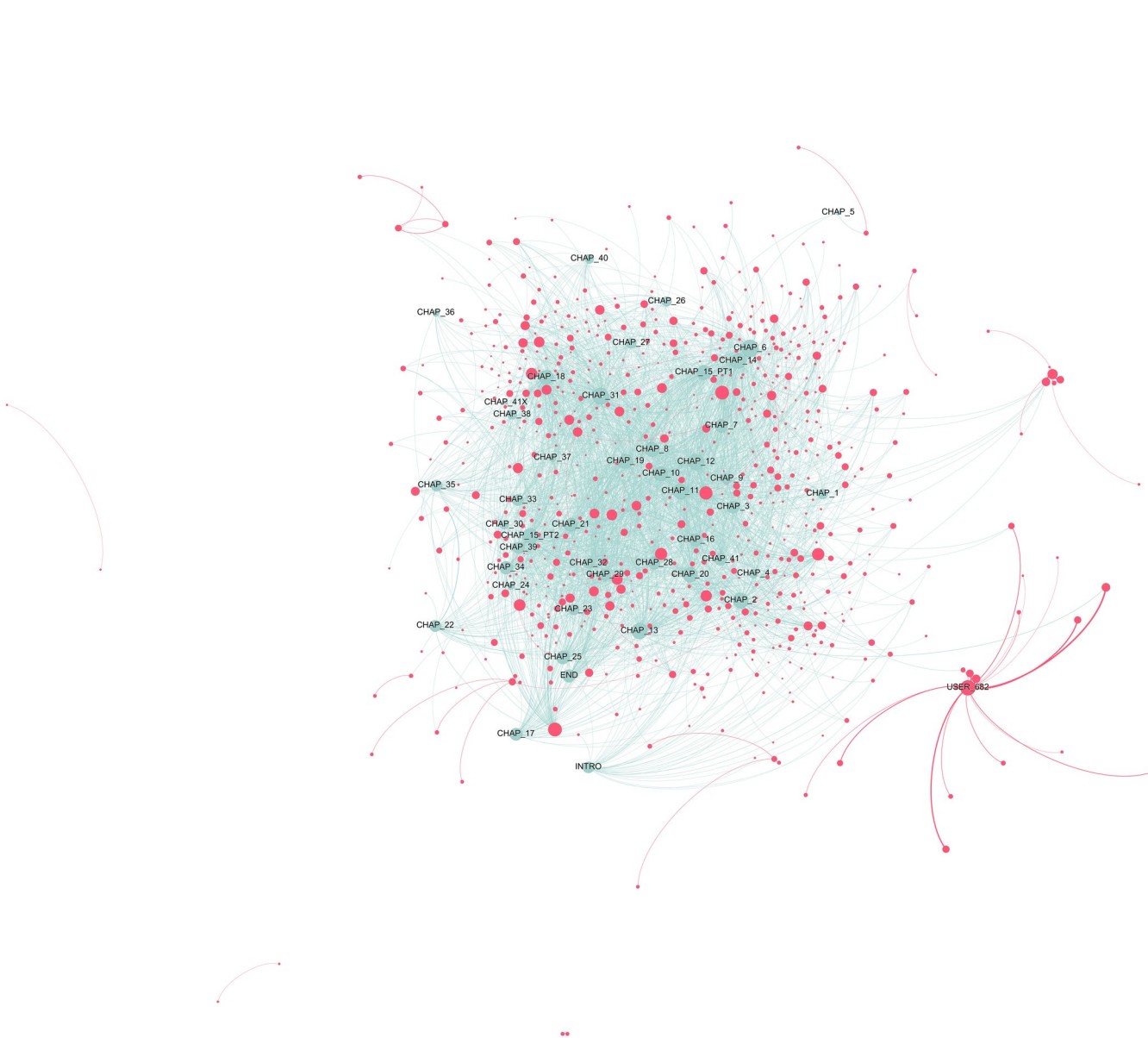

**Fig 11. Users-chapters network graph for *The Bad Boy's Girl*.** The full graph contains 735 nodes, 0.6% of the total in the book (130,615). This reduction was obtained by taking into consideration only nodes that interacted between each other at least 30 times. Weighted degree of the nodes (i.e. their dimension) is visualized through a logarithmic scale, thus half diameter corresponds to 1/10 of the weighted degree. We chose this option to better highlight distinctions. Due to the large dimension of the network, the quality of the image is limited. A full-scale figure, as well as an interactive Gephi file are available as supporting information (S7 Fig and S5 File).

nerves. They are my old friends. I have heard you mention them with consideration these last twenty years at least", the most recurrent comments are "savage" and "slay", referring to the sharpness of Mr. Bennet's utterance. Something similar also happens in chapter 3. The two most commented paragraphs of the chapter are the ones in which Mr. Darcy is presented for

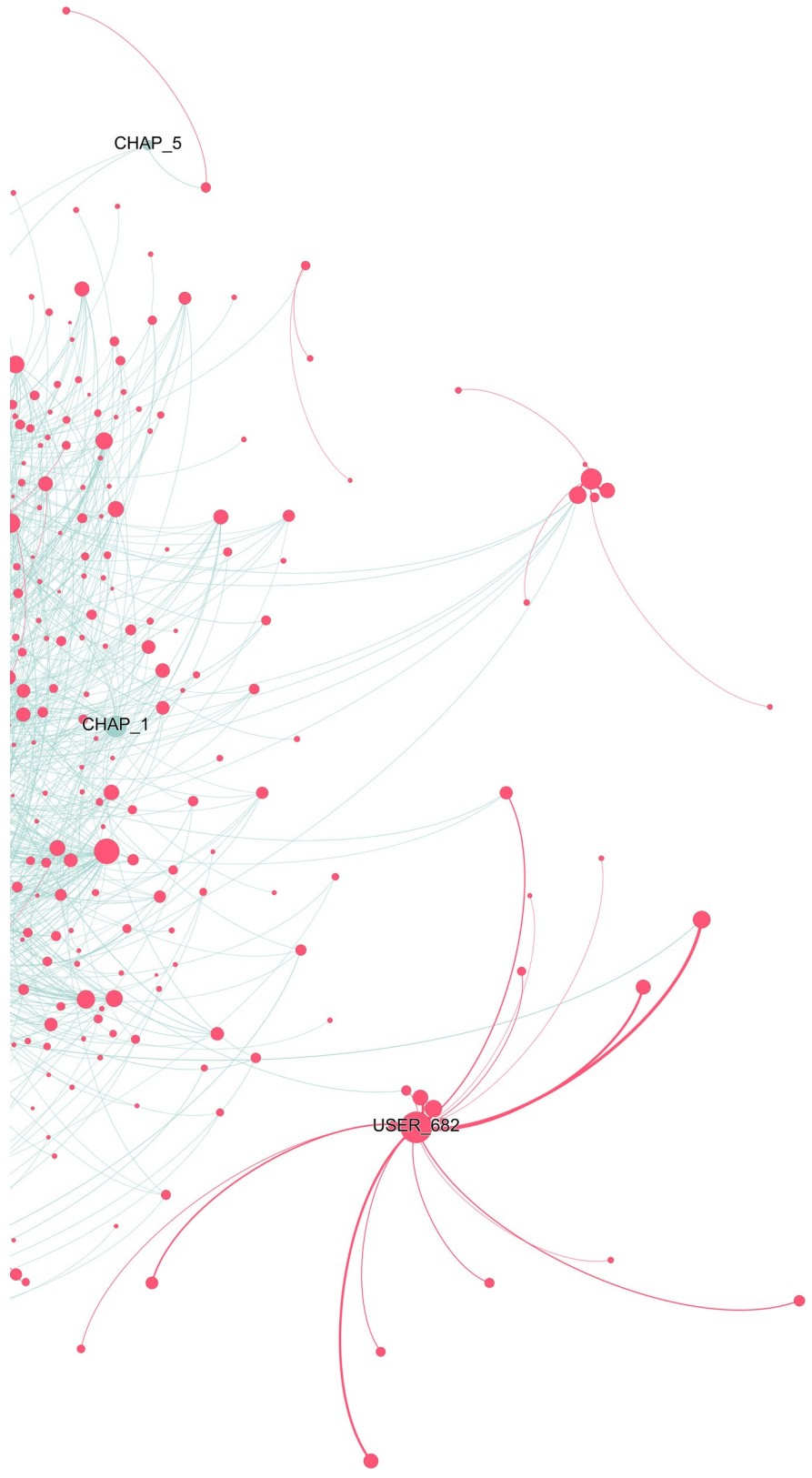

**Fig 12. Users-chapters network graph for *The Bad Boy's Girl* (detail).** The full graph contains 735 nodes, 0.6% of the total in the book (130,615). This reduction was obtained by taking into consideration only nodes that interacted

between each other at least 30 times. Weighted degree of the nodes (i.e. their dimension) is visualized through a logarithmic scale, thus half diameter corresponds to 1/10 of the weighted degree. We chose this option to better highlight distinctions.

the first time and the one in which—talking to Mr. Bingley—he dismisses Elizabeth as "tolerable, but not handsome enough to tempt *me*". Readers' comments are mostly reactions to his arrogance and rudeness in saying such things openly.

For TF, Wattpad's social dimension plays a more important role in eliciting comments than for Classics. The number of comments to the two overall most commented paragraphs is at least four times bigger than the number of comments to other paragraphs. As many stories on Wattpad, *Growing Up* is a serialized novel, chapter 38 is one of the last of the book and its final paragraphs are an "author's note" asking readers with whom of two boys they would like the female character to have a relationship. The plot denouement is still undecided and the author is giving agency to their readers by asking them what their desires are. However, beside this moment the other paragraphs of *Growing Up* did not elicit more than 6,000 comments each. On the contrary, *She's With me* is a story able to trigger more than 4,000 comments for 50 different paragraphs (see dataset). The most commented is at the beginning of the story (ch. 1, par. 42): readers just had a glimpse of the main character's thoughts and in this paragraph it is the first time she speaks in the story. She accidentally bumps into the school's "bad boy" and, after his rude and offensive words, she utters a ferocious and hilarious reply that amazes all the students in the corridor. The other most commented paragraphs are a mystery about the protagonist's past (ch.1, par. 7, 9,500 comments), a surprising behaviour of the bad boy (ch. 13, par. 106), the revelation that both boys like her and are jealous (ch. 38, par. 37), the girl's admission that she is horny (ch. 54, par. 78, 9,400 comments), and the first kiss after a very long suspense (ch. 56, 111, 9,400 comments). Other highly commented paragraphs are *My Wattpad Love* scene in which one of the two boys finally says to the girl "You're mine" (ch. 28, par. 174), and the long-awaited kiss of *The Hoodie Girl* (ch. 44, par. 160).

*The Bad Boy's Girl* has 15 paragraphs with more than 4,000 comments, all in different chapters. Chapter 31 has two of them: one is the anticipatory title—"What It Feels Like To Get Your Heart Broken"—where readers are very emotionally expressing their concern for what they are about to read, fearing that Tessa and Cole might break up; the other one is the last paragraph of the chapter, just after the cliffhanging cheating confession, where Tessa thinks "This is what it feels like to get your heart broken and smashed right? If it is, then why do people even bother falling in love?". It is the most tragic moment in the story and the character's feelings are expressed with two questions. In the comments, we see readers answering these

**Table 6. Most commented paragraphs of the categories Classics and Teen Fiction.**

| | Classics | Total comments | | Teen Fiction | |
|---|---|---|---|---|---|
| 1 | *Pride and Prejudice* (ch. 1, par. 1) | 2,600 | 59,600 | *Growing Up* (ch. 38, par. 158) | 1 |
| 2 | *Wuthering Heights* (ch. 1, par. 1) | 1,400 | 57,700 | *Growing Up* (ch. 38, par. 159) | 2 |
| 3 | *Romeo and Juliet* (Prologue, par. 2) | 484 | 15,300 | *She's With Me* (ch. 1, par. 42) | 3 |
| 4 | *Pride and Prejudice* (ch. 1, par. 29) | 428 | 13,200 | *I Sold Myself to the Devil* (ch. 4, par. 32) | 4 |
| 5 | *Pride and Prejudice* (ch. 1, par. 26) | 400 | 12,600 | *Stay With Me* (ch. 23, par. 68) | 5 |
| 6 | *Pride and Prejudice* (ch. 3, par. 5) | 381 | 11,300 | *She's With Me* (ch. 13, par. 106) | 6 |
| 7 | *Pride and Prejudice* (ch. 3, par. 13) | 372 | 11,200 | *My Wattpad Love* (ch. 28, par. 174) | 7 |
| 8 | *Romeo and Juliet* (Act 1, scene 1, par. 24) | 364 | 10,800 | *The Hoodie* (ch. 44, par. 160) | 8 |
| 9 | *Pride and Prejudice* (ch. 1, par. 2) | 337 | 10,000 | *She's With Me* (ch. 13, par. 106) | 9 |
| 10 | *Pride and Prejudice* (ch. 1, par. 14) | 333 | 9,600 | *She's With Me* (ch. 38, par. 37) | 10 |

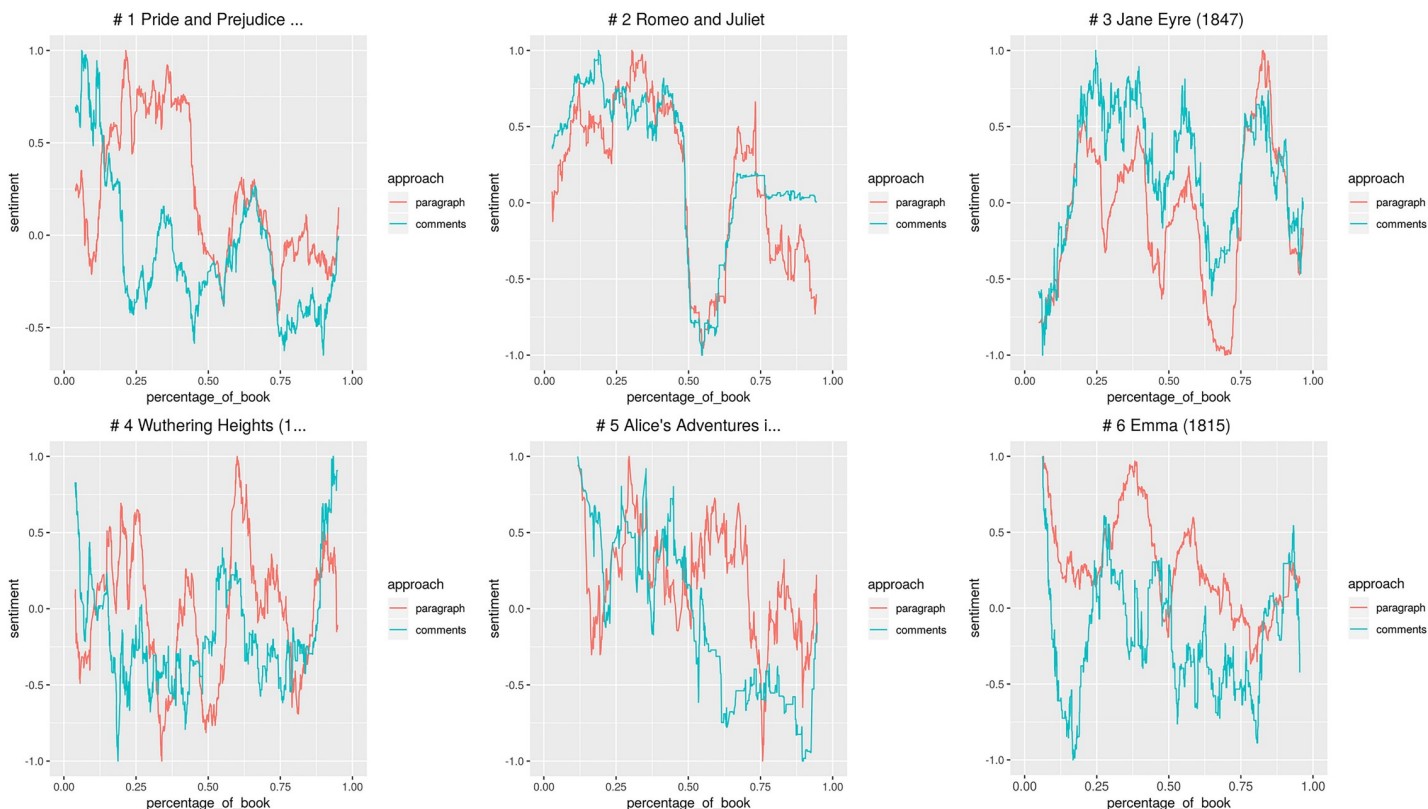

**Fig 13. Graphs with the emotional arcs of story and comments for 6 Teen Fiction stories.** All individual graphs are available as supporting information (S8–S13 Figs).

questions, as if they were addressed directly to them, or verbalizing their emotional embodied distress for the sad event. For instance, more than 500 people say they are crying (dataset), and many others share their similar personal experiences. Other highly commented paragraphs in the novel are surprising story events (ch. 24, par. 89; ch. 26, par. 102), a scene that readers already anticipated in previous chapters (ch. 22, par. 55), honest and hilarious characters' thoughts or utterances (ch. 4, par. 56; ch. 8, par. 46; ch. 12, par. 151; ch. 30, par. 20), a reference to a very famous song (ch. 13, par. 112), a revelation about Cole's past (ch. 14, par. 8), and a narrative ellipsis that denies to readers satisfaction after a long perceived suspense (ch. 41, par. 86).

After having browsed the most commented paragraphs, we used sentiment analysis to compare the emotional valence of the story and the comments of each of the 12 books (Figs 13 and 14).

From a first look at the graphs we can spot a major difference between the two genres: the two arcs have much more harmonious trends for TF than for Classics. We also checked (1) correlations between the sentiment values of the two emotional arcs and (2) effects of story's emotional valences on readers' emotional responses. We found significant correlations for all 12 titles ($p < .001$), with higher values for TF (Table 7). *Pride and Prejudice* is the title with the lowest correlation (0.11). We also performed linear regressions to check whether the emotional effect of the stories on readers' response varied between the two genres. All TF stories have a positive effect higher than 0.5, with *She's With Me* reaching 0.99 ($R^2 = 0.62$). Classics have a much lower effect, the lowest being *Pride and Prejudice* (0.13, $R^2 = 0.01$) (Table 8). *Romeo and Juliet* (0.81, $R^2 = 0.75$) and *Jane Eyre* (0.72, $R^2 = 0.64$) are exceptions. We will discuss the reasons behind these values in section 4.5.

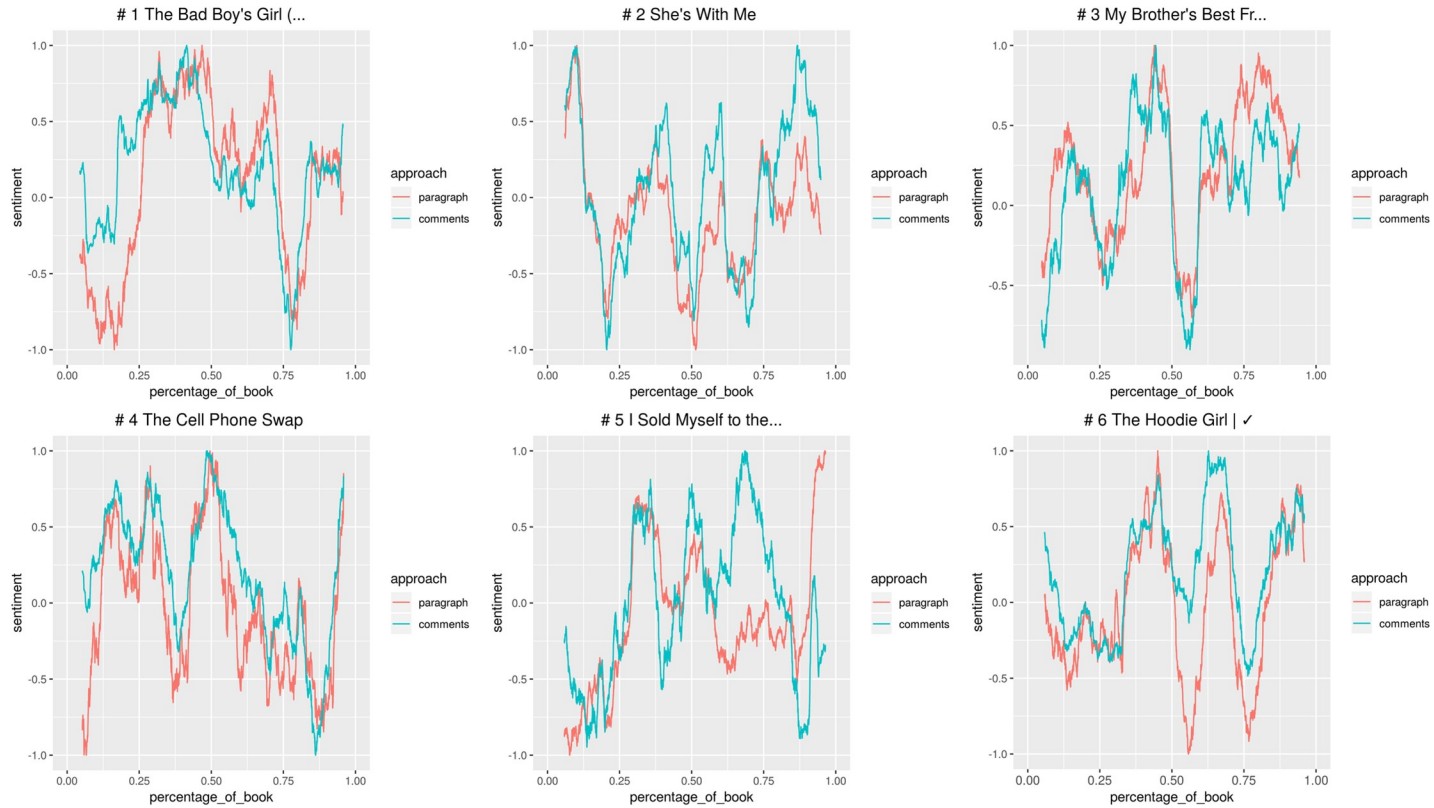

**Fig 14. Graphs with the emotional arcs of story and comments for 6 Classics.** All individual graphs are available as supporting information (S14–S19 Figs).

To better understand the relation between the emotional valence of story and comments, we focused on *Pride and Prejudice* and *The Bad Boy's Girl*, selecting for close reading the parts with bigger gaps between story and comments or extreme values in either of them. Fig 15 shows the emotional arcs of *Pride and Prejudice* with highlighted the parts that we analysed.

For *Pride and Prejudice*, the first notable interval regards chapters 7 to 10 (range 6.3–12.7%), from Jane's departure to visit Mr. Bingley at Netherfield, where she falls sick, until the

**Table 7. Results of the analyses of correlation between the sentiments of story and comments.**

| Book | Pearson r | p | t | Degrees of freedom | Confidence interval | |
|---|---|---|---|---|---|---|
| | | | | | lower limit | upper limit |
| *Pride and Prejudice* | 0.11 | 1.333e-06 | 4.85 | 1848 | 0.067 | 0.157 |
| *Romeo and Juliet* | 0.87 | < 2.2e-16 | 72.92 | 1762 | 0.854 | 0.878 |
| *Jane Eyre* | 0.80 | < 2.2e-16 | 80.13 | 3642 | 0.787 | 0.810 |
| *Wuthering Heights* | 0.23 | < 2.2e-16 | 9.98 | 1714 | 0.189 | 0.278 |
| *Alice's Adventures in Wonderland* | 0.40 | < 2.2e-16 | 11.59 | 692 | 0.339 | 0.463 |
| *Emma* | 0.33 | < 2.2e-16 | 15.78 | 2074 | 0.288 | 0.365 |
| *The Bad Boy's Girl* | 0.69 | < 2.2e-16 | 71.14 | 5516 | 0.678 | 0.705 |
| *She's With Me* | 0.79 | < 2.2e-16 | 79.56 | 3828 | 0.777 | 0.801 |
| *My Brother's Best Friend* | 0.74 | < 2.2e-16 | 100.63 | 8391 | 0.730 | 0.749 |
| *The Cell Phone Swap* | 0.84 | < 2.2e-16 | 114.90 | 5365 | 0.835 | 0.851 |
| *I Sold Myself to the Devil for Vinyls . . . Pitiful I Know* | 0.47 | < 2.2e-16 | 67.50 | 15730 | 0.461 | 0.486 |
| *The Hoodie Girl* | 0.80 | < 2.2e-16 | 98.86 | 5359 | 0.794 | 0.813 |

**Table 8. Results of the linear regressions to estimate the effect of the sentiment of the story on the sentiment of comments.**

| Book | b | Std. error | p | t | Degrees of freedom | F | R² |
|---|---|---|---|---|---|---|---|
| *Pride and Prejudice* | 0.13 | 0.027 | 1.33e-06 | 4.85 | 1848 | 23.53 | 0.012 |
| *Romeo and Juliet* | 0.81 | 0.011 | < 2.2e-16 | 72.92 | 1762 | 5317 | 0.751 |
| *Jane Eyre* | 0.72 | 0.009 | < 2.2e-16 | 80.13 | 3642 | 6421 | 0.638 |
| *Wuthering Heights* | 0.21 | 0.021 | < 2.2e-16 | 9.98 | 1714 | 99.55 | 0.055 |
| *Alice's Adventures in Wonderland* | 0.56 | 0.048 | < 2.2e-16 | 11.59 | 692 | 134.2 | 0.162 |
| *Emma* | 0.40 | 0.026 | < 2.2e-16 | 15.78 | 2074 | 248.9 | 0.107 |
| *The Bad Boy's Girl* | 0.52 | 0.007 | < 2.2e-16 | 71.14 | 5516 | 5062 | 0.478 |
| *She's With Me* | 0.99 | 0.012 | < 2.2e-16 | 79.56 | 3828 | 6331 | 0.623 |
| *My Brother's Best Friend* | 0.77 | 0.008 | < 2.2e-16 | 100.60 | 8391 | 10130 | 0.547 |
| *The Cell Phone Swap* | 0.79 | 0.007 | < 2.2e-16 | 114.90 | 5365 | 13200 | 0.711 |
| *I Sold Myself to the Devil for Vinyls . . . Pitiful I Know* | 0.53 | 0.008 | < 2.2e-16 | 67.50 | 15730 | 4557 | 0.225 |
| *The Hoodie Girl* | 0.66 | 0.007 | < 2.2e-16 | 98.86 | 5359 | 9773 | 0.64 |

moment Mr. Darcy is writing a letter to her sister, under the gaze of Miss Bingley. The paragraphs have an almost neutral average sentiment (+0.3/-0.1), whereas comments peak (+0.5/+1) before a long declining slope begins. Many readers especially "like" and "love" Mr. Bennet's frank remarks about his younger and "silliest" daughters. Then come the dialogues in chapter 8, Mr. Bingley's and Mr. Darcy's replies to Miss Bingley in defence of Elizabeth and Jane: "I thought Miss Elizabeth Bennet looked remarkably well when she came into the room

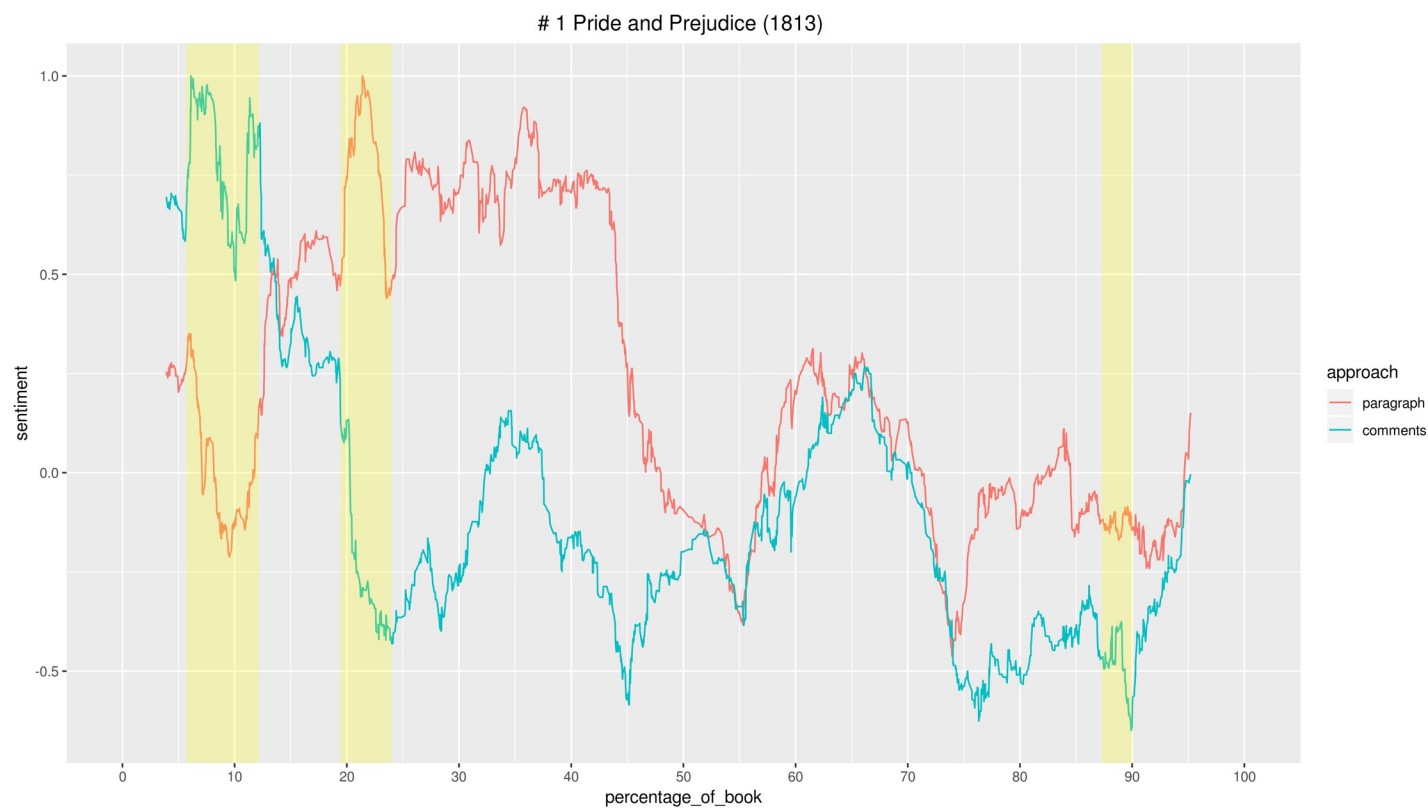

**Fig 15. Graph with emotional arcs for *Pride and Prejudice* text and comments.**

this morning. Her dirty petticoat quite escaped my notice", "It shows an affection for her sister that is very pleasing", "they [Elizabeth's eyes] were brightened by the exercise", and "If they had uncles enough to fill all Cheapside, . . . it would not make them one jot less agreeable". But also Elizabeth's reply to Miss Bingley's insinuations: "'Miss Eliza Bennet', said Miss Bingley, 'despises cards. She is a great reader, and has no pleasure in anything else'. 'I deserve neither such praise nor such censure', cried Elizabeth; 'I am not a great reader, and I have pleasure in many things'". Readers also appreciate Elizabeth's comment on Mr. Darcy's too demanding idea of an "accomplished woman": "I am no longer surprised at your knowing only six accomplished women. I rather wonder now at your knowing any". In chapters 9 and 10, it is again Elizabeth's way of talking and Mr. Darcy cold attitude towards the obviously flirting Miss Bingley that amuse readers. They also start mocking her, generating a series of hilarious social interactions about funny comments. About the sentiment attribution process to the comments in this interval, we have to point out that the positive value of "like" in the dictionary is misleading, since in the comments it mostly occurs as either a preposition or a conjunction, not as a verb.

The second notable interval includes the scenes from Mr. Wickham revelation of his dispute with Mr. Darcy, until the dialogue between Elizabeth and Mr. Darcy at the ball at Netherfield (ch. 16, par. 21 –ch. 18, par. 39; range 19.4–23.5%). This is the interval with the biggest sentiment gap between paragraphs' highest positive peak (+0.5/+1) and comments' negative slope (+0.2/-0.4). In the novel, Mr. Collins's exaggerated courtesy, Mr. Bennet teasing him, Elizabeth's positive impression of the good-looking Mr. Wickham, the invitation to the ball, and the politeness displayed by everybody during the event, all contribute to the positive sentiment. Looking closely at the words used for sentiment attribution and comparing them with the text, we noted that positive words like "flattered", "civilities", "vivacity", "acquaintance" and "grace" are not in the Syuzhet dictionary, but they occur in the paragraphs with the highest positive values (range 21.1–22%). In the comments, "bad" occurs often to describe Mr. Wickham's way of talking about Mr. Darcy, an act that irritates readers. "Annoying" and "bitch" refer to Mr. Wickham and the excessively eloquent Mr. Collins—who makes his first appearance here— especially when he praises Lady Catherine. "Hate" is all for Mr. Wickham, since he is in the way of the love between Elizabeth and Mr. Darcy. "Prejudice" is mentioned often because this is the part where most readers realize that Elizabeth is prone to judge very quickly, based on a first-impression. "Rude" is for Mr. and Miss Bingley, who bring the invitation to the ball to the Bennets but leave very quickly, avoiding Mrs. Bennet. Interestingly, readers express this while admitting that they would do the same, since they dislike Mrs. Bennet.

The third notable interval is close to the end of the novel, when Mr. Bingley proposes to Jane and a merry time follows in the Bennets' house (ch. 55; range 88.1–90%). Here paragraphs have a slightly negative sentiment (-0.1/-0.2), while comments reach a negative peak (-0.3/-0.6). We noticed that the negative sentiment of the novel is affected by the occurrence of the word "cried" in many paragraphs, which is considered negative by Syuzhet but it is actually a verb commonly used by Jane Austen to refer characters' speech. Similarly, "enough" is never used in a negative way in this segment. Moreover, the sentiment dictionary does not include words like "cordiality", "perfections", "satisfaction", "satisfy", "handsomest", and the British-spelled "favour" and "favourite". In the comments, we see that readers do not seem very involved with Jane's engagement, they rather see it as the awaited step that can eventually allow Elizabeth to marry Mr. Darcy. The negative peak corresponds to Jane's line: "I am certainly the most fortunate creature that ever existed! . . . Oh! Lizzy, why am I thus singled from my family, and blessed above them all! If I could but see you as happy! If there were but such another man for you!" Readers comment on this with sentences like "wait" and "don't worry", which increase the negative sentiment. However, in the comments to this chapter the highly

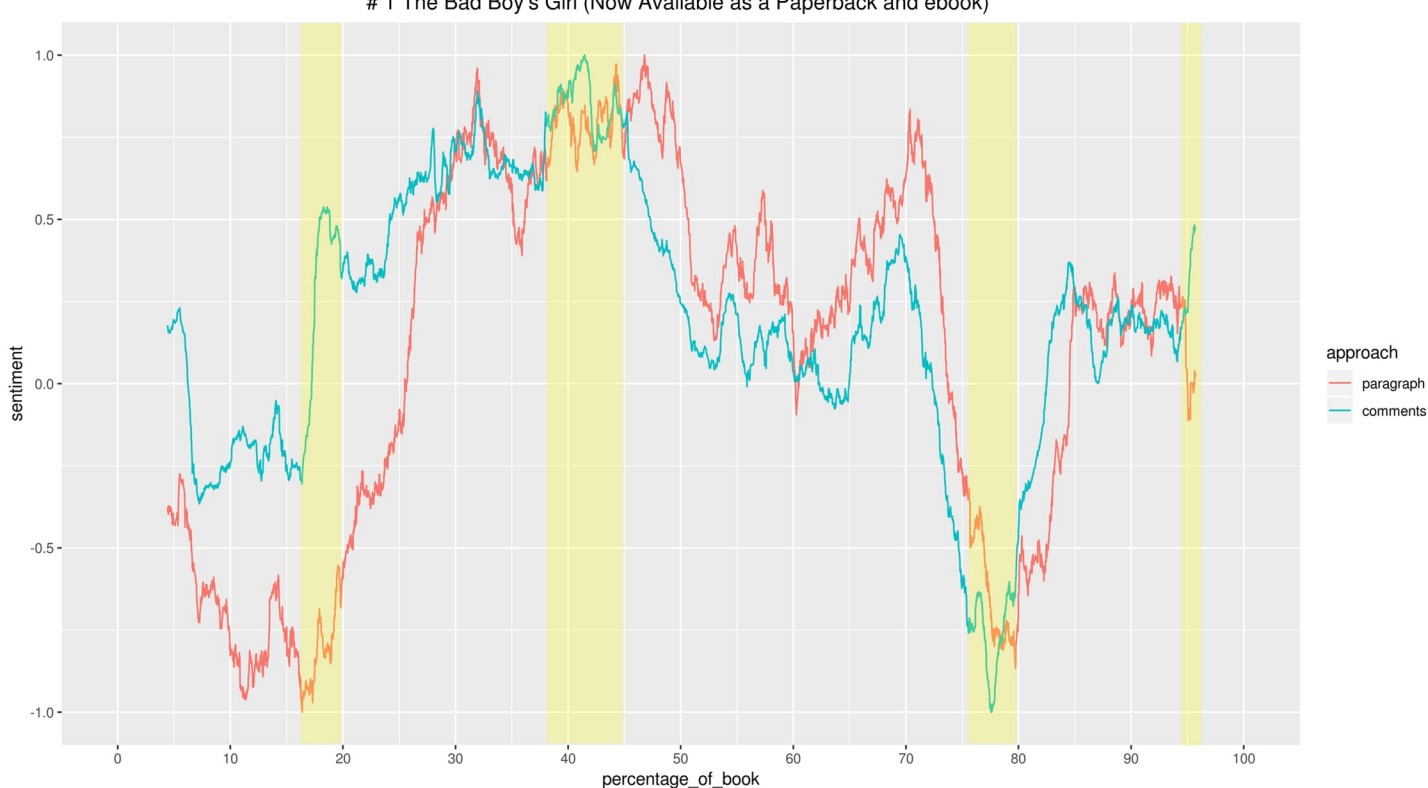

**Fig 16. Graph with emotional arcs for _The Bad Boy's Girl_ text and comments.**

negative words used by Syuzhet to calculate the mean sentiment of the segment do not occur. "No", "bitch", "damn" and "slay" occur abundantly in comments to the beginning of the next chapter, when Lady Catherine appears and talks to Elizabeth. As an effect of the moving window, the comments negative peak is assigned with a 1% shift.

Fig 16 shows the emotional arcs of _The Bad Boy's Girl_ with highlighted the parts that we analysed. The first notable interval (ch. 8, par. 34—ch. 9; range 16.2–19.8%) corresponds to the events after the party where Tessa has been assaulted. At first she is at home with Cole and the Jay arrives, then she is chatting with her best friends Beth and Megan at school (range 18– 19%), also talking about her fight with Cole on the night of the party. In the next scene she is "kidnapped" into the janitor's closet by a "stranger". Paragraphs' sentiment reaches here a negative peak (-1/0.6), while comments' sentiment is rising to positive values (-0.3/+0.5). In the novel, Tessa is conflicted: she likes Cole more and more and is trying to avoid falling for him, reminding to herself that he is a "bad guy" and that people can get "hurt" staying around him. This is the part when the conflict between the two brothers—Cole and Jay—manifests openly ("hell", "mad", "anger"). Interestingly, the negative peak (range 16.2–17.2%) is actually attributed to the intimate moment between Tessa and Cole, which precedes the argument between the brothers: they are playfully fighting and eventually jump in the swimming pool, where there is physical sexual tension for the first time. As it happened with _Pride and Prejudice_, the peak is slightly shifted due the effect of the moving window. Comments' sentiment has a positive peak at 18.6%, the scene of the conversation with Beth and Megan. Tessa thinking about matching Beth and Cole—i.e. the "bad girl" and the "bad boy"—sparks a great discussion, with a lot of mentions of high-score words like "good" and "bad". Moreover, readers say they like

Beth and that she is a good friend, but the positive sentiment is also supported by the context, with incidental mentions of details in the preceding paragraphs that triggers excited responses: readers "love" indie music, Shania Twain, Edgar Allan Poe, and Barbies.

The second notable interval includes chapters 14 to 18 (range 38.3–45%). Paragraphs are very high (+0.6/+0.95) and comments have a peak (+0.7/+1) before they start declining. The results for this interval are quite straightforward. Tessa is becoming aware that she is falling in love with Cole and she is excited by seeing him half-naked. Cole's "cute" side is shown: he is a "softie" with his grandmother and asks Tessa to go and visit her together. The comments reflect this positive part of the narration. Most of them are literally just the word "love" or "cute". Many re-readers affirm that they "love" this part of the book. It is worth noting that, as already seen elsewhere, "like" is very often used either as a preposition or a conjunction, especially in the novel. Likewise, "right" is used as an adverb or in an interrogative way, either in Tessa's interior monologue or when her brother Travis is asking about her recent life.

Even the third notable interval (ch. 32–33; range 75.6–80%) is pretty straightforward to interpret. It is a deep declining slope for both paragraphs (-0.3/-0.7) and comments (-0.7/-1). These are the chapters after Cole's confession that he cheated on Tessa. She is in deep sorrow and this is reflected in her dramatic thoughts. He is also suffering for what he did and tries to apologize. There are also a few scenes in which there are yells and harsh words. The comments reflect both the pain and the hatred felt by the characters. "Stupid" and "bitch" are for Cole, for the girl with whom he cheated, and for Tessa because she cannot see that Cole still loves her. "Hate" is for Cole but also for Tessa because she cannot forgive him. And some readers also hate other readers because they are "haters" who do not understand how love relationships go. "Crying" has a high negative value in this interval, used both by readers, who describe themselves in such a state, and in the text, because Cole is also crying and showing his remorse.

The last notable interval is towards the end (ch. 41; range 94.4–95.7%), where the sentiment of paragraphs (+0.2/-0.1) and comments (+0.2/+0.5) starts to diverge. In the previous chapter, Tessa's mother has expressed some doubts about her relationship with Cole. In this chapter, Tessa's father talks to them about whether their relationship is "bad", "wrong" or a "problem", but also about their mother having to "stop" interfering. There is "tension" and both Tessa and her father look "angry". From the comments we can see that the father's speech starts a discussion about what it means to "love" someone and readers sadly admit that he is "right" and has a "good" point. On the level of narration, readers "love" the anticipating title "BAM, You're Naked and It's Go Time", and they beg "please" the author for this to be the chapter in which Tessa and Cole have sex. Similar remarks appear also in comments to other chapters' titles, whose meaning—after actually having read the chapter—is often surprisingly revealed to be different from what was expected.

## 4. Discussion

### 4.1 Corpus

Reports about reading often rely on market data [83,84], some surveys collect data from randomized samples of the population asking directly about reading habits [85,86], but only a few of them take into account the changing behaviour of young readers including specific questions about reading non-standard book formats like Wattpad stories [87]. The results in such reports are consistent worldwide: reading is an activity in strong decline after childhood [85,88–92]. The magnitude and the steady increase of the activity happening on Wattpad contradicts these data. This is a sign that more appropriate methodology to report about reading needs to be developed, in order to take into account the changes of societies in which digital technology is widespread and influences reading habits.

With more than 30 million stories, Wattpad is the biggest database of user-generated fiction, surpassing both *FanFiction.net* and *Archive of Our Own* [93,94]. English is clearly the dominant language, representing nearly 80% of the corpus (Table 1), but there are important collections of stories written in non-Western languages, as well. There are more Turkish stories than French ones, and more Vietnamese and Indonesian stories than German ones. Such a wide and diversified database of story titles, with links to access the texts and the comments, is an extremely useful resource for the investigation of both literary production worldwide and the communication and social dynamics of teenagers sharing a passion and talking about their lives. The corpus of titles is complete but the metadata subcorpus is still limited to English Classics and Teen Fiction stories and thus requires an extension that will also include other languages and genres/categories.

## 4.2 Stories

If "world literature" is conceived as an "interactive space" that involves the contribution of both authors and readers [95], Wattpad is the ideal context where to carry out an analysis of the most recent evolution of world literature. While globalization is gradually disrupting the closed canons of traditional literary criticism [96], recent studies have demonstrated how, in wide collaborative projects such as Wikipedia, the conception of world literature has already changed significantly [97]. With Wattpad, the scenario evolved even further: new generations of "wreaders" [98] are not simply expanding canonical narrative worlds (fanfiction and Classics are only a small portion of Wattpad stories), but are actually taking over as original creators, by sharing the most widespread international language: that of narrative invention. In our study, we have just touched the surface of the phenomenon, by demonstrating already how wide and far-reaching a categorization of Wattpad stories can be, and how much further, in-depth studies will be needed in the immediate future. For instance, to understand why English-speaking authors do not face absence (the word "without" rarely occurs in titles), or why only Western authors are concerned with the creation of new worlds and with the relation between characters and world, whereas Asian authors are focusing more on family relationships. This distinction confirms what claimed by psychological studies about the different construction of the "Self" in Western and Oriental cultures, with more references to family relationships in blogs written by Asians [99,100].

Analysing the most frequent words occurring in titles is a very simple way to explore the themes of stories that proved to be effective in highlighting similarities and differences between literatures across the world (Tables 2, 3 and 4). It was a big surprise to acknowledge that the main cultural influence in Wattpad's world literature does not come from Western prestigious literary models but from a small Asian country's music industry, South Korean K-Pop. This is true also for Western literatures, marking a significant change in comparison to past literary traditions. Reading authors' notes, it is clear that teenagers who publish on Wattpad aspire to become famous authors, but the model they choose to follow is not that of literary elitism. They rather embrace in full the social dimension of Wattpad, choosing popular music references and engaging in continuous conversations with the readers [28]. Further investigation will need to focus on the evolution of themes year after year and on the relation between frequent themes and most popular stories.

## 4.3 Readers

Reader response is something that changes according to the reader's experiential background, her language, country of origin, and culture. By mapping Wattpad's users we can have an idea about books' reception by looking at the cultural specificity of the readers we are studying:

among the readers of a selection of English language stories, only part of them is natively anglophone (Fig 1). There is a great variety of geographical locations, even though the texts are only in English. The extent of the mismatch between the language of texts and users' location shows that there are many people reading English stories in original language, although they live in countries in which English is not the main language. Part of them may be native English-speaking migrants living in foreign countries, but there may be also people reading in English as a second language. Filipino readership is particularly big: they not only write and read in their native language (Table 1), they also read a lot in English, confirming that they are one of the most relevant groups among Wattpad readers [101].

Cultural studies have diversified and relativized the perspectives we take when we investigate reader response, contesting the dominance of white voices in literary studies [102,103]. Wattpad is evidence that in the 21st century, thanks to digital technology, diversity in readers' response can also happen simultaneously on the same reading platform. It is important to acknowledge that English-speaking Wattpad readers are spread across the world and they are engaging in a social exchange with people in many different countries. This is an opportunity to meet other perspectives on books—and how they affect people's lives—by starting a dialogue about differences and similarities in how stories are received by readers with different cultures that share the same passion. The prosocial effect that fiction has [104–106] is complemented by the opportunity of actual social interaction offered by digital social reading practices. Further research will need to assess whether social reading is indeed enhancing the prosocial effect of fiction.

Another important aspect to underline is that Wattpad readers' construction of identity occurs exclusively through the text of their comments, unlike face to face reading groups, and it is affected by the public social context. Both the fact of taking place on a social-network-like platform and of participating in a reading group strengthen the performative aspect of identity construction [9]. We did not analyse reader response on Wattpad from this perspective, but comments in which users say whether they are rereaders or first-time readers is an example of this phenomenon. The influence of social space on how readers present themselves is not something happening on 21st century social reading only, it also has a historical antecedent in notes by famous readers. In fact, it has been noted that some of John Keats's annotations in the margins of books have "the signs of performance or at least of a consciousness of shared experience about it" [107].

## 4.4 Engagement

By analysing how Wattpad readers engage with Classics and Teen Fiction, we highlighted that there are different kinds of readers using Wattpad, and that the social aspect of reading and genre is affecting the volume of the commenting activity. Classics have a concentration of many comments in the first chapters, whereas TF stories are widely commented until the end (Figs 2 and 3). Reading some random comments for different titles, we saw that one of the reasons is the difficulty to understand the language in which Classics are written, but further research is needed to determine how important this factor is. Indeed, the low average of comments to Classics (Table 5), probably means that users abandoned the book after reading a few paragraphs or chapters, without commenting much. This is also confirmed by the network graph of *Pride and Prejudice*, which shows an isolated cluster of users around the first chapters (Fig 9).

Readers of TF have a sustained engagement throughout the stories and the most commented chapters are sometimes in the middle or towards the end of the books (Fig 3). But to have an increase in the number of comments, the number of readers should likely remain the

same for the whole story (see dataset, for the number of readers for each chapter). One of the reasons this is happening for TF but not for Classics is that the former are mostly serialized novels, that is chapters are published online as soon as the author has written them, usually with a weekly pace, but the frequency of update can vary widely. This feuilleton-like publishing style likely affects readers' engagement, since there is a suspense effect generated by paratextual elements, which cannot be present for Classics, since they were uploaded in their complete form. Interestingly, something similar happened with those Classics that were originally published in a serialised form, appearing weekly as newspaper supplements. For instance, there is written evidence of readers' response in the form of letters to the newspapers that were publishing serial novels in the 19th century in the UK [108].

Looking at the most commented paragraphs (section 3.5), we saw that peaks in the number of comments are related to story events that have a strong narrative interest, i.e. related to suspense, curiosity, or surprise effects [109]. This is in line with one of the main functions of highlighting, as shown by Rowberry [110] on a corpus of Kindle Popular Highlights: users highlight pivotal narrative moments. However, in highly commented paragraphs there is no trace of the commonplaces identified by Rowberry: moral values do not spark many discussions among Wattpad readers, although this might be due to genre-specific limitations of our corpus or to the readers' young age. More generally, highlighting and commenting seem to have two distinct functions, even though both activities are publicly shared and not meant for individual use only. Similarly, Barnett [4] noted that more prestigious books have fewer Kindle highlights and notes than popular and free books, also prompting different kinds of behaviours: the former have notes reflecting attentive and critical reading, the latter have a lot of chatting.

A representation of the flow of comments of the most active users also confirmed that genre affects the commenting activity on Wattpad. Readers that commented both Classics and TF are more active on TF titles but also read and commented Classics (Fig 5). This suggests that encouraging teenagers to read TF on Wattpad might help them to develop a passion for reading Classic literature as well. Looking at the most active users, we observed their interest in Classics was only focused on *Pride and Prejudice*, *Jane Eyre*, and *Wuthering Heights*, suggesting that these three books have something that makes them close to TF stories. Reading the comments we realized that one of the reasons is that these Classics entered the cultural universe of Wattpad readers being mentioned in popular TF stories. Being a generic famous story, like *Alice's Adventures in Wonderland*—which has been adapted and transformed in different audiovisual formats (films, video games, comics, etc.)—is not enough, as attested by the almost absolute disregard by TF readers for this title (Fig 5 and Table 5). There is also another kind of readers that reads mostly Classics but did not seem to be interested in TF (Fig 4). However, we cannot exclude that they engaged with other genres on Wattpad. Further research will be needed to understand what other reading preferences characterise readers of Classics on Wattpad.

Network analysis showed another difference between popular and prestigious literature: TF readers have fewer social interactions than readers of Classics. The difference is 8.9%, but the TF network graph (Fig 8) exaggerates this gap because we set to 250 interactions the lower limit for nodes to be visualised. This means that all users who in total had less than 250 interactions are not visualised. Despite this deformation, the visualisation was useful to highlight that TF is able to prompt intense social interactions—i.e. clustering of users—more than Classics. The TF and Classics network graphs have to be compared considering that we had to reduce the number of nodes visualized by setting a threshold of minimum number of interactions which is different for the two categories (5 for Classics, 250 for TF). There were more user-user interactions for Classics (30.6% vs. 21.7% for TF) but readers of TF generated groups with

stronger bonds, that is readers that are able to interact with other readers more than 250 times (Fig 8). This happened both when users were reading the same book and across different books. Indeed, the network graphs show that the novels have a large amount of readers in common and, therefore, are kept closer in the network. Further explorations of the network may focus on the interactions within cliques, i.e. users who form groups within which everybody is connected to everybody else. This will allow to investigate what motivate readers to have intense conversations while reading.

Further investigation is also needed to clarify whether and how textual features contribute to this remarkable difference between genres. So far, this aspect has remained unexplored even by other researchers who use network analysis for the study of social reading, like Chiara Faggiolani and Lorenzo Verna, who analysed the Italian platform *aNobii* focusing on the relations between books and reading preferences [111,112]. The fact that their data are not accessible does not allow us to compare whether in different social reading platforms there are similar relationships between popular and prestigious literature.

The results of this literary modelling implemented combining different techniques brings about differences and similarities in how people engage with reading two different genres socially, but they can be used to plan further in-depth qualitative analysis of notable users. For instance, to investigate the motivations of the high sociability of USER_2411 in reading TF books (Fig 8) and of USER_277 (Fig 10) with *Pride and Prejudice*, but also the intense commenting of users E and Q (Figs 4 and 5). Within the scope of the present work, some insights about the reasons for the difference between the social activity with Classics and TF came from analysing the emotional valence of stories and readers' response, and from reading the content of the comments and the corresponding paragraphs. We discuss this in the next section, but first we have to underline that the genre-related comparison we wanted to do is actually more complex: Teen Fiction and Classics are different not only because of the popular/prestige distinction they represent, but also for other oppositions: 19th/21st century, mature/teenage authors, North-American/British authors, whole-novels/serialized-novels. We considered all these differences in interpreting the results of our analyses and future research will have to be designed taking them into account.

## 4.5 Emotions

Looking at the most commented paragraphs, we discovered that readers like when characters react to violence, bullies, and rude people. Not really in a physical way, rather with sarcasm and words that can shut up the abuser. Similarly, candid characters trigger positive comments, whether they are outspoken or it is only their thoughts that are reported. In the latter case, it remains to be investigated whether the narratorial voice and the point of view affect readers' response [113,114]. Appreciation for such characters' actions occurs both in comments to TF and Classics: utterances by Elizabeth, Mr. Darcy, Mr. Bingley and Mr. Bennet, but also Amelia (*She's With Me*), Beth, and Tessa's thoughts (*The Bad Boy's Girl*). Of course, this kind of positive reactions is possible only if there is a negative character that previously acted or talked in an offensive or annoying way: Mrs. Bennet, Mr. Collins, Miss Bingley, Lady Catherine, but also Mr. Darcy in some dialogues with Elizabeth. And for TF, the "bad boys" and the "queen bees", Aiden (*She's With Me*), Jay, Cole, Nicole (*The Bad Boy's Girl*). For Classics, it is interesting to note that commenting on Wattpad is different than highlighting on a Kindle e-readers. In the latter case, Rowberry [110] has shown that readers focus on "perceived wisdom" and "affective climaxes of a narrative". We have shown that the most commented paragraphs on Wattpad display more basic narrative effects—suspense, curiosity and surprise—not necessarily related to emotionally intense scenes. The high number of comments on the first

paragraphs also happens to be an implicit sign of emotional engagement, since some readers claim that they are rereading the story, plausibly because they felt involved during their first reading. Accordingly, there is a subset of comments written by rereaders, which we did not analyse in detail. Interestingly, this is a practice similar to the cumulation of annotations written in different times in the margins of printed books [107], although comments on Wattpad seem to be mainly addressed to other readers, sometimes even warning them when there may be spoilers. We also showed one example of another relevant aspect of TF, the interactive nature of serial publishing on Wattpad—and fanfiction websites in general [115]—which activates readers' interest to participate in the story construction, manifesting their emotional engagement with characters and expressing their desired plot development.

Thanks to sentiment analysis we were able to quantify and compare the emotional response of a large number of readers. We confirmed that in many cases the emotional valence of a story has a direct effect on reader response. This result is not trivial because the verbalisation of emotional response to a story can take many forms and it is not obvious that positive words in stories trigger readers' positive utterances. This is an interesting phenomenon that requires further investigation to understand the relation between textual features and reader response. Previous research using sentiment analysis on online reviews showed that they are characterised by "intimacy", inasmuch as the expression of emotions is one of the main feature of the verbalisation of reader response online [116,117]. However, the correlation with other kinds of textual features is still unclear.

Based on our analysis, we can claim that, among the texts we considered, *Pride and Prejudice* has the lowest correlation and effect probably because of the many user-user interactions (30.3% of all interactions): readers engage more in conversation and their comments reflect this shift from a direct emotional response to the text towards a discussion about the book—a more social and cognitive-oriented kind of activity. The results will have to be confirmed analysing the networks and the sentiment of other literary works, but this is already an important insight for educators. The resistance of young readers to reading novels written in an English difficult to understand can be overcome by peer learning and collective intelligence [37,118,119], as attested by the reciprocal help we witnessed in the comments, where users explain each other paragraphs that they did not understand. In this regard, one of the most remarkable examples is that of users who paraphrased the most difficult paragraphs of *Pride and Prejudice* in the comments, receiving the praise and thanks of many other readers. A similar form of collective intelligence also emerges from interactions on fanfiction websites, where readers collaborate in figuring out plot developments but also details about the story setting [120].

With respect to the innovative methodology that we applied, we gained some valuable feedback. Concerning the attribution of sentiment value to comments, we noted that positive response did not correspond to positive sentiment only. For instance, it is true that many comments reacting to witty characters mention "yes", "well said", "I'm enjoying this", "I love her", but there are also many enthusiastic expressions like "burn", "slay", "savage", "damn", which are considered as negative sentiment. Syuzhet default dictionary is not suited to correctly interpret neither this kind of comments nor the lingo used by the Wattpad community (e.g., "lmao" for "laugh my ass out", "lol" for "laugh out loud", "cinnamon roll" for "cute, sweet", etc.). As these examples show, an important thing to remember is that negative sentiment does not necessarily mean that reader response is negative. A specific sentiment dictionary will need to be prepared in order to assign appropriate values to these expressions.

With respect to sentiment attribution to novels, Syuzhet dictionary showed some flaws when applied to a 19th century British English text, since it does not include important words or is not correctly tuned for the historical use of some words. On the other hand, although

Syuzhet dictionary only includes a limited sample of words, it nevertheless identified the correct sentiment of paragraphs with the highest positive values without taking many other positive words into account. From a stylistic point of view, this result can be seen as a sign that positive words "keep each other company". From a methodological point of view, it can be a sign that Syuzhet dictionary is good enough for the purpose of identifying sentiment values in novels, provided that the sentiment dictionary is tuned correctly beforehand. On TF novels the default dictionary works better, probably because it has been created on the basis of six 21st century American novels, including one that can be considered Teen Fiction [74]. Further research is also needed to confirm whether the fact that for *Romeo and Juliet* and *Jane Eyre* the sentiment of the novel has a strong effect on the sentiment of comments is actually an exception, or rather it is the sentiment of the other Classics that has been misattributed. With respect to this, a comparison with other literary genres may be helpful.

Another methodological remark concerns the use of the moving window, because it can be questioned whether it is a good option. Certainly, it can help to identify trends in extended parts of text, without focusing only on the effect of single paragraphs. This is positive because reading is an activity following a linear progression and the sentiment of previous paragraphs affects how we read the following paragraphs. However, using this method we have to be careful when interpreting the results, as we showed in the case of *Pride and Prejudice*, where the comments' most negative sentiment does not actually correspond to the exact part of the text to which it is linked (the R Markdown file explaining how Syuzhet calculates the relevance of single words in generating a section of the sentiment analysis graph can be found in the supplementary information S1 File). If the aim is to more precisely identify the emotional valence of specific textual or narrative features, it may be more fruitful to interpret the raw sentiment values. In this way, the sentiment value will be calculated only based on the words occurring in the selected part of text. After the computer has precisely identified sentiment values, it will then be easier to employ human expertise to interpret what narrative or stylistic phenomena are related to them. However, in this way it is more difficult to contextualize the selected text in a trend of wider scope (e.g., declining or rising sentiment, harmonious or divergent sentiment between text and comments). It should also be considered that the moving window helps to correct the mistakes that are frequently generated when performing sentiment analysis on short texts, such as Wattpad's paragraphs and comments. Even the most advanced techniques—combining sentiment dictionaries with syntactic parsing and deep learning—struggle to reach an acceptable efficiency [121], especially when dealing with historical texts [122].

Regarding our third hypothesis, we can affirm that comparing the sentiment of stories and comments provided empirical evidence to link textual features to readers' emotional response. From the analysis of matching and diverging intervals we got confirmation that witty characters are very much appreciated, as already suggested by the most commented paragraphs. Moreover, having secondary characters whose personality create a contrast with that of the protagonists is a strategy that amplifies readers' reaction. For instance, Miss Bingley with respect to Elizabeth, and the double contrast between the proud but warm-hearted Mr. Darcy and the charming but manipulative Mr. Wickham. A similar process can also work for a single character, like in the case of Tessa's continuous internal conflict: first, between the attraction for Cole and the resistance she has because of her childhood memories; second, between the love she feels for Cole and the will to keep him at a distance after he hurt her. Comments show that readers mimic a similar conflict, especially after an extreme inversion of sentiment in the story (ch. 32): they feel for Tessa and hate Cole because he made her suffer, but they also hate Tessa for not forgiving him and empathize with Cole.

Another textual feature that can warm up emotional response is the mention of details not directly related to the plot that help to prompt identification and empathy with the characters.

For instance, the incidental mention of Beth's love for indie music or Tessa reading Edgar Allan Poe and growing up playing with Barbies (ch. 9). Lastly, titles that generate expectations which are later surprisingly disattended can be used to spark interest and discussions among readers.

## 5.Conclusions

We presented Wattpad as a resource for literary studies, claiming that neglecting its role in the contemporary reading landscape will inevitably lead to misleading assumptions about reading behaviours, especially of teenagers. We tested three hypotheses and brought evidence to support our arguments: 1. analysing the themes of around 30 million stories published on Wattpad we highlighted differences and similarities of reading interests across 13 different languages, contributing to the study of world literature; 2. analysing users' behaviours in reading Classics and Teen Fiction we showed that readers' engagement on Wattpad changes according to the literary prestige of texts; 3. comparing the sentiment of stories and comments we provided empirical evidence to link textual features to readers' emotional response.

We need to be careful in generalizing about readers' behaviours, since the texts and the readers on Wattpad are only a subgroup of all kind of literary texts and readers. The young age of Wattpad users and their still relative small cultural capital may affect the way they talk about books, as it also happens in other kinds of digital platforms [8,123,124]. Moreover, it should be kept in mind that we did not focus on the response of individual readers, we rather created a statistical model of a collective response to stories.

By examining reader response at scale we can set the ground for statistically valid claims about how a lot of people read. Namely, we can understand what kind of emotions readers feel towards the characters and how they react to specific narrative strategies like suspense. Some of the comments we presented in this article show that readers' responses are often contradictory, even for short paragraphs, and there is not just one kind of reader [10,125]. Ultimately, the comments are a resource for different kinds of inquiries, investigating how during reading: values are negotiated; identities are constructed; authors interact with readers; metaphors are interpreted; intertextual connections are built; factors like "experiential background" [126] and "personal relevance" [127] affect reader response; etc. It has been debated whether marginalia can actually help to understand reading practices of the past [7,128–132]. In this regard, studying Wattpad comments to understand contemporary reading practices has the advantage that the retrieved information can be compared with other kinds of data within more comprehensive reading research programmes [58]. For instance, the quantitative analysis of comments can be used to identify readers that commented a lot and invite them for interviews about their reading experience.

Social reading is rising to prominence among all the various reading practices, namely thanks to the use of digital media. Looking ahead, we have to acknowledge that Wattpad readers are the generation of new readers. A generation who is nurturing a passion for reading being immersed in the use of digital media. Pessimistic claims about the effect of reading on screen compared to paper [133–136] will have to be reconsidered taking into account the changed social context in which millions of reading acts are taking place.

Analysing reading in the digital age can also be useful to understand social processes of reading in general: there is not one reading habit, but many ways to read. Here we showed the differences between reading Classics and Teen Fiction. Our results can be particularly useful for educators, who can find here some indications about the actual reception of Classics by teenagers and develop new approaches to foster engagement with literature.

## Supporting information

**S1 File. R Markdown files describing the computer-assisted analyses.**
(ZIP)

**S2 File. Gephi file for the Classics users-books network.**
(GEPHI)

**S3 File. Gephi file for the TF users-books network.**
(GEPHI)

**S4 File. Gephi file for *Pride and Prejudice* users-chapters network.**
(GEPHI)

**S5 File. Gephi file for *The Bad Boy's Girl* users-chapters network.**
(GEPHI)

**S1 Fig. Interactive map of users' location extracted from a corpus of 12 English language stories (*n* = 35,208).**
(HTML)

**S2 Fig. Interactive graph of the flow of comments for readers who wrote at least 200 comments to Classics.**
(HTML)

**S3 Fig. Interactive graph of the flow of comments for readers who wrote at least 100 comments to both Classics and TF.**
(HTML)

**S4 Fig. High-resolution image for the Classics users-books network.**
(PNG)

**S5 Fig. High-resolution image for the TF users-books network.**
(PNG)

**S6 Fig. High-resolution image for *Pride and Prejudice* users-chapters network.**
(PNG)

**S7 Fig. High-resolution image for *The Bad Boy's Girl* users-chapters network.**
(PNG)

**S8 Fig. Graph with emotional arcs for *The Bad Boy's Girl* text and comments.**
(TIF)

**S9 Fig. Graph with emotional arcs for *She's With Me* text and comments.**
(TIF)

**S10 Fig. Graph with emotional arcs for *My Brother's Best Friend* text and comments.**
(TIF)

**S11 Fig. Graph with emotional arcs for *The Cell Phone Swap* text and comments.**
(TIF)

**S12 Fig. Graph with emotional arcs for *I Sold Myself to the Devil for Vinyls. . . Pitiful I Know* text and comments.**
(TIF)

**S13 Fig. Graph with emotional arcs for *The Hoodie Girl* text and comments.**
(TIF)

**S14 Fig. Graph with emotional arcs for *Pride and Prejudice* text and comments.**
(TIF)

**S15 Fig. Graph with emotional arcs for *Romeo and Juliet* text and comments.**
(TIF)

**S16 Fig. Graph with emotional arcs for *Jane Eyre* text and comments.**
(TIF)

**S17 Fig. Graph with emotional arcs for *Wuthering Heights* text and comments.**
(TIF)

**S18 Fig. Graph with emotional arcs for *Alice's Adventures in Wonderland* text and comments.**
(TIF)

**S19 Fig. Graph with emotional arcs for *Emma* text and comments.**
(TIF)

## Acknowledgments

The authors would like to thank Erzsébeth Tóth-Czifra from Open Methods for her suggestions regarding text mining and copyright, and Anna Al-Damluji from Flourish for her support during the preparation of data visualisations.

## Author Contributions

**Conceptualization:** Federico Pianzola, Simone Rebora, Gerhard Lauer.

**Data curation:** Federico Pianzola, Simone Rebora.

**Formal analysis:** Federico Pianzola, Simone Rebora.

**Funding acquisition:** Federico Pianzola.

**Methodology:** Federico Pianzola, Simone Rebora.

**Visualization:** Federico Pianzola, Simone Rebora.

**Writing – original draft:** Federico Pianzola, Simone Rebora, Gerhard Lauer.

**Writing – review & editing:** Federico Pianzola, Simone Rebora, Gerhard Lauer.

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
