## [Decision Letter · Decision Letter 0]

26 Jul 2019

PONE-D-19-16904

Wattpad as a resource for literary studies in the 21st century: Quantitative and qualitative examples of the importance of digital social reading and readers’ comments in the margins

PLOS ONE

Dear Dr. Pianzola,

Thank you for submitting your manuscript to PLOS ONE. After careful consideration, we feel that it has merit but does not fully meet PLOS ONE’s publication criteria as it currently stands. Therefore, we invite you to submit a revised version of the manuscript that addresses the points raised during the review process.

Many thanks again for submitting your paper. The reviewers agree, in general, that the article needs to be reworked and revised before publication. Please take a look at the detailed reviewers comments, I hope they will be self-explanatory, and have the article proof-read.

We would appreciate receiving your revised manuscript by Sep 09 2019 11:59PM. To enhance the reproducibility of your results, we recommend that if applicable you deposit your laboratory protocols in protocols.io, where a protocol can be assigned its own identifier (DOI) such that it can be cited independently in the future. For instructions see: http://journals.plos.org/plosone/s/submission-guidelines#loc-laboratory-protocols

We look forward to receiving your revised manuscript.

Kind regards,

David Orrego-Carmona, Ph.D.

Academic Editor

PLOS ONE

**Journal Requirements:**

2. Thank you for clarifying that you obtained permission from Wattpad to scrape their data and that you consulted with various bodies about the legal and ethical dimension of your work. Could you please in the revision: a) state in your methods section that you did indeed obtain permission from Wattpad to scrape the data; b) in your ethics statement, explain who you consulted, how you protected users' privacy and what other measures you took to comply with ethical standards. Thank you very much in advance.

3. In order to meet the requirements for the Science of Stories collection, the Guest Editors ask that you please make the code to reproduce your analysis available in a stable, public repository (for example, Zenodo, or GitHub) or a suitable cloud computing service (such as Code Ocean) when submitting your revised manuscript. The code should include a license file and detailed readme so that someone with access to the dataset is able to reproduce your analysis using the code. We ask that you include the DOI for the repository holding your code in an updated Data Availability statement with your revised manuscript.

4. We note that  Figure(s) 1 in your submission contain [map/satellite] images which may be copyrighted. All PLOS content is published under the Creative Commons Attribution License (CC BY 4.0), which means that the manuscript, images, and Supporting Information files will be freely available online, and any third party is permitted to access, download, copy, distribute, and use these materials in any way, even commercially, with proper attribution. For these reasons, we cannot publish previously copyrighted maps or satellite images created using proprietary data, such as Google software (Google Maps, Street View, and Earth). For more information, see our copyright guidelines: http://journals.plos.org/plosone/s/licenses-and-copyright.

a) You may seek permission from the original copyright holder of Figure(s) [#] to publish the content specifically under the CC BY 4.0 license.  

**Comments to the Author**

1. Is the manuscript technically sound, and do the data support the conclusions?

Reviewer #1: Yes

Reviewer #2: Yes

Reviewer #3: Yes

2. Has the statistical analysis been performed appropriately and rigorously? 

Reviewer #1: Yes

Reviewer #2: Yes

Reviewer #3: Yes

3. Have the authors made all data underlying the findings in their manuscript fully available?

Reviewer #1: Yes

Reviewer #2: Yes

Reviewer #3: Yes

4. Is the manuscript presented in an intelligible fashion and written in standard English?

Reviewer #1: Yes

Reviewer #2: Yes

Reviewer #3: No

5. Review Comments to the Author

Reviewer #1: This manuscript offers an interesting and innovative study of text and paratext on Wattpad in terms of social reading and sentiment analysis. While there have been previous studies of Wattpad, this is, to my knowledge, the most extensive, and will most certainly provoke further discussion of methods for studying Wattpad, as well as broader debates around digital reading communities and contemporary book history.

The manuscript is well structured and highly readable. There are detailed explanations of techniques used. All relevant datasets and R workbooks have been included in the manuscript. The interactive visualisations enhance the reader's comprehension of the primary argument. The majority of analysis is robust and provides sufficient evidence for the proposed conclusions. There is a good mixture of appropriate data analysis techniques to triangulate the paper's main hypotheses. The demographic data presents the greatest challenge. The underlying point of demonstrating international appeal is clear, but how can you ensure the data is reliable. For example, I note 2 users in North Korea, which is as likely to be soft trolling as genuine users.

The authors integrate a broad range of scholarship from relevant disciplines, and demonstrate they are keeping on top of even the most recent publications. At times, the scholarship is underplayed to make claims for originality unnecessarily ("We show for the first time...", p.1), but otherwise this is clearly building on previous work in an exciting way. There are some instances in the bibliography where references require a bit more information to ensure that the material can be located.

My only major concern with the manuscript comes from the ethics of data mining reader data at scale. To my mind, the challenges of researching social media at scale should be addressed in the methodology or in an ethics statement (which would presumably fall under 'human subject research'). While individuals are offered pseudonyms, this may not be sufficient protection, particularly since the authors acknowledge that "data of the comments corpus cannot be shared publicly because this will expose underage users to possible identification." These are some of the trickiest issues around studying reading in 'live' online environments, and at a bare minimum it would be useful to know that this has been reviewed by the authors' institutional ethics board, but preferably this would also include further discussion in the manuscript itself.

A few further comments that might be addressed in a further revision:

p. 14 - the claim that 'thanks' should be treated as a negative in the corpus would benefit from further evidence.

p. 41-2 the focus of the manuscript is on comments, but do likes/views across chapters confirm that most readers abandon the texts? These would be more useful metrics as they require a lower threshold for engagement.

Despite some of my concerns, the manuscript will be a much needed intervention in studies of reading online.

Reviewer #2: ‘Wattpad as a Resource for Literary Studies in the 21st Century’

Decision: Accept with major revisions

Topic:

The submission is a study of internet-hosted social reading examining readerly interactions about both digital-born teen fiction and established print classics on the globally-dominant Canadian-based platform Wattpad. The cross-linguistic nature of the sample and the researchers’ proficiency in a range of languages are benefits given Wattpad’s international userbase. The study rightly acknowledges that this userbase skews young, female and English-speaking (although not necessarily native-speaking) and thus that the study results cannot be generalised to all online reading formations.

The chief limitation of the study is the way its near-exclusive focus on texts and reader comments brackets off considerations of production, and even of the wider import of the issues raised. In the submission’s late stages many avenues for future research are flagged but these involve a near-exclusive focus on analysing the existing or additional datasets, rather than broader issues of theoretical or commercial import. In short, the submission needs to look up from the text and data occasionally.

Theory:

The submission lacks contextualisation of its research in reader response theory since the 1970s (classically Iser, but also landmark 1980s works such as Fish and Radway). N.B. especially recent work emerging from book history and publishing studies on readers’ uses of digital platforms (DeNel Rehberg Sedo and Danielle Fuller, Ann Steiner, Anouk Lang’s edited collection). The first of these explicitly discusses reading groups’ cross-cultural interpretations of Pride and Prejudice and hence seems highly relevant here. Lisa Nakamura’s piece for PMLA on Goodreads is also worth mentioning, especially in light of the below methodological concerns).

Overall, the theoretical coverage feels thin compared to the methods section.

Method:

The submission makes innovative use of ‘distant reading’ computer-assisted methods for contemporary text corpora. As the authors note, the methodology is almost exclusively used for historical corpora. The mixed methods approach blending quantitative big-picture and qualitative close-reading analysis is logical and well-justified.

The paper is exhaustive in its social sciences-style detailing of the project’s research methodology, corpus development and data visualization techniques. Some of the material in section 3.4 seems repetitious, adumbrating minor variations in datasets or visualisations. It created a strong impatience in this humanities-trained reader to get to the actual analysis (largely delayed until p. 38’s ‘Discussion’ section). Even then, the paper suffers from a lopsidedness, with its vast methodological apparatus generating insufficient analytical pay-off or theoretical takeaways.

One effect of the social-sciences focus on quantification and method is that methodologies that cast their net wider than the text itself are overlooked. e.g. the political economy of Wattpad and its commercial attempts to distil some archetypal story structure from its vast archive to drive print publication and screen adaptation deals (see https://www.theglobeandmail.com/business/article-what-clicks-with-fiction-readers-wattpad-helps-authors-find-out/ ). If such approaches are not going to be engaged with, they should at a minimum be registered and reasons for not exploring them provided.

Literature review:

The submission evidences a broad and multidisciplinary familiarity with extant research. The ‘lament for reading’ school (Baron et al.) makes a late entrance on p.49 but key texts are mentioned. Sven Birkerts’ is the other name commonly found in this company.

The first Simon Peter Rowberry reference [29] is incomplete. His article about Kindle Highlights would seem highly relevant but is not mentioned until p.45. See also Bronwen Thomas’ articles on fan fiction and textual interactivity, as well as her forthcoming monograph on social media and literature.

Analysis:

The relative paucity of readers responding to other readers (less than a third of all comments) creates the suspicion that Wattpad is a potentially interactive reading space, but that much of the actual reader commenting serves a more performative function (cf. lines1057-58). Lines 737-38 appear to reinforce this suspicion, though it is not examined in any detail. Implications of this should be teased out and tested against social reading theory relating to, for example, Goodreads.

There is no escaping the reality that most of the reader comments cited are notable mostly for their banality (e.g. character judgements about Pride and Prejudice). A later comment that a certain passage in a teen fiction ‘sparks a great discussion’ bases this estimation on the number of high-sentiment words it involves, not on the interpretive rigour or ingenuity of such reader comments. This raises the question of whether amateur readers like Wattpad’s users are more worth studying than expert readers such as literary critics, whose professional skills elucidate ‘hidden’ or underexamined innovative readings of existing texts. What exactly do the researchers aim to elucidate by examining amateur readers’ responses at scale if it is not the content of their comments? Granted, high-school English teachers might take heart that digital platforms create reader engagement with classic texts, even in antiquated forms of English, but would any teacher award high marks for the levels of readerly comprehension and analysis quoted in the submission?

The authors’ point about the ‘feuilleton-like’ serial publishing format driving commenting frequency for teen fiction is legitimate, but overlooks the fact that many 19thC classic Anglo-American works were published serially, if not the Bronte and Austen works most commented upon in Wattpad (cf. Dickens, Thackeray).

Expression:

There are pervasive minor ESL slips (e.g. lines 65-66, 232, 758-59, 927, 956, 1098-99, 1136, 1140, 1148, 1222, 1226, 1269) and lapses in noun-verb agreement (e.g. line 1064, 1100, 1156, 1214), preposition choice (line 381, 569-70, 658, 776) and possessive apostrophes (1027).

Non-grammatical sentences occasionally also occur (lines 338-39).

‘Literary studies’ is usually handled as a singular noun

In summary, the submission would benefit from lifting its gaze from the specifics of the study methodology and the Wattpad texts themselves to analyse more of the production and consumption dimensions of Wattpad, specifically the firm’s monetisation of readerly affect, and the theoretical implications (and limitations) of analysing amateur readers’ digital interactions.

17 July 2019

Reviewer #3: This is an important contribution to the scholarship. The submission performs distant and close reading of the texts in a way that expands knowledge and its methodologies. The findings are very useful and open up new knowledge about reading in the digital age.

The findings are relevant and based on good interpretation of the data.

Although I acknowledge that there is a significant body of literature referred to here, it seems to me not always the right or relevant literature. For example, there is a whole field of study around manuscript annotations that should be referred to beyond Kerby-Fulton et al (2012). Heather Jackson's Marginalia, for example. Bonnie Mak's work could be useful here too.

Similarly, reader response theory is a broad field that requires more than a single reference to Bortolussi and Dixon (2003) - an empirical study - to avoid misrepresenting the field.

I suggest Katherine Bode's work in Reading by Numbers and A World of Fiction. And, hesitatingly, I also point out my own article "Social Reading: The Kindle’s Social Highlighting Function and Emerging Reading Practices" just for interest in particular because it discusses readers commenting on Pride and Prejudice, though within the Kindle platform rather than the Wattpad platform.

Terminology and definitions

- social reading isn't limited to online spaces (though it is facilitated by them) and the two shouldn't be conflated.

- the authors use 'literary stories' but I think they mean fiction (rather than getting into the debate about what is and isn't literary)

I don't know of the use of stop words as a verbal phrase as it is used here (line 219). As far as I am aware, it is a noun phrase that would need another verb to make the sentence grammatically correct.

Further errors in expression:

Line 46 - our established literary studies (perhaps should be material literary collections?)

54-5 - readers' responses

65 - Although Wattpad is in place since more than a decade...

78 already existing research intended

There are many examples like this throughout the article and this is why I indicated expression problems at question 4 but these will be easily fixed by a proofreading.

6. PLOS authors have the option to publish the peer review history of their article (what does this mean?). If published, this will include your full peer review and any attached files.

Reviewer #1: No

Reviewer #2: No

Reviewer #3: Yes: Tully Barnett

---

## [Author Response · Author response to Decision Letter 0]

13 Sep 2019

All responses have been included in the attached file "Response to reviewers"

---

## [Decision Letter · Decision Letter 1]

4 Nov 2019

PONE-D-19-16904R1

Wattpad as a resource for literary studies. Quantitative and qualitative examples of the importance of digital social reading and readers’ comments in the margins

PLOS ONE

Dear Dr. Pianzola,

Thank you for submitting your manuscript to PLOS ONE. After careful consideration, we feel that it has merit but does not fully meet PLOS ONE’s publication criteria as it currently stands. Therefore, we invite you to submit a revised version of the manuscript that addresses the points raised during the review process.

We would appreciate receiving your revised manuscript by Dec 19 2019 11:59PM. To enhance the reproducibility of your results, we recommend that if applicable you deposit your laboratory protocols in protocols.io, where a protocol can be assigned its own identifier (DOI) such that it can be cited independently in the future. For instructions see: http://journals.plos.org/plosone/s/submission-guidelines#loc-laboratory-protocols

We look forward to receiving your revised manuscript.

Kind regards,

David Orrego-Carmona, Ph.D.

Academic Editor

PLOS ONE

Reviewers' comments:

Reviewer's Responses to Questions

**Comments to the Author**

1. If the authors have adequately addressed your comments raised in a previous round of review and you feel that this manuscript is now acceptable for publication, you may indicate that here to bypass the “Comments to the Author” section, enter your conflict of interest statement in the “Confidential to Editor” section, and submit your "Accept" recommendation.

Reviewer #1: All comments have been addressed

Reviewer #3: All comments have been addressed

2. Is the manuscript technically sound, and do the data support the conclusions?

Reviewer #1: Yes

Reviewer #3: Yes

3. Has the statistical analysis been performed appropriately and rigorously? 

Reviewer #1: Yes

Reviewer #3: Yes

4. Have the authors made all data underlying the findings in their manuscript fully available?

Reviewer #1: Yes

Reviewer #3: Yes

5. Is the manuscript presented in an intelligible fashion and written in standard English?

Reviewer #1: Yes

Reviewer #3: Yes

6. Review Comments to the Author

Reviewer #1: The revised manuscript is much stronger and addresses all my initial concerns or the authors' comments provide appropriate justification for not making amendments.

Reviewer #3: This article represents an important contribution to the scholarship on reading in digital environments and the use of quantitative methodologies to illuminate new aspects of the reading process and reader response. The manuscript has been improved through the revision process.

I have two further questions the authors might like to consider:

Where is the evidence that wattpad stories are *written* on mobile phones?

What role might platform recommendations have played (ie if you read this you might like to read this book) in associations between texts?

In addition, the manuscript will require some more editing for English expression. For example, the less/fewer error, reader response vs readers' response (sometimes one or the other is required)

7. PLOS authors have the option to publish the peer review history of their article (what does this mean?). If published, this will include your full peer review and any attached files.

Reviewer #1: No

Reviewer #3: No

---

## [Author Response · Author response to Decision Letter 1]

28 Nov 2019

response to the reviewers has been uploaded as a separate file

---

## [Editor Report · Decision Letter 2]

5 Dec 2019

Wattpad as a resource for literary studies. Quantitative and qualitative examples of the importance of digital social reading and readers’ comments in the margins

PONE-D-19-16904R2

Dear Dr. Pianzola,

We are pleased to inform you that your manuscript has been judged scientifically suitable for publication and will be formally accepted for publication once it complies with all outstanding technical requirements.

With kind regards,

David Orrego-Carmona, Ph.D.

Academic Editor

PLOS ONE
---

## [Editor Report · Acceptance letter]

16 Dec 2019

PONE-D-19-16904R2 

Wattpad as a resource for literary studies. Quantitative and qualitative examples of the importance of digital social reading and readers’ comments in the margins 

Dear Dr. Pianzola:

I am pleased to inform you that your manuscript has been deemed suitable for publication in PLOS ONE. Congratulations! Your manuscript is now with our production department. 

With kind regards,

on behalf of

Dr. David Orrego-Carmona 

Academic Editor

PLOS ONE